# Soil carbon debt from land use change in Brazil

João M. Villela [1], Júnior M. Damian [2], Daniel R. P. Gonçalves [3], Luis G. Barioni [2], Maurício R. Cherubin [1,4] & Carlos E. P. Cerri [1,4] ✉

Carbon farming is a fundamental strategy for mitigating climate change. Brazil, with 276 million hectares of agricultural land, has strong potential to lead this agenda, but uncertainty about soil carbon (C) debt hinders understanding of its true mitigation capacity. Here, we estimate the soil carbon gap, the difference between soil organic carbon (SOC) stocks under native vegetation and agricultural land across Brazil's six biomes, which represents the theoretical potential for soil recarbonization. A meta-analysis using a comprehensive national SOC database (4,290 records, 0–30 cm) is used to estimate an overall carbon debt of $1.40 \pm 0.1$ Pg C. The results show that sustainable practices such as crop rotation, intercropping, no-tillage, and integrated agricultural systems enhance SOC recovery. These findings highlight Brazil's capacity to drive global emissions mitigation, guide low-carbon policies, and position the country as a key actor in the emerging global carbon market.

Quantifying the soil carbon (C) gap resulting from land-use change is fundamental for understanding the magnitude of human-induced depletion and the potential for soil carbon recovery[1,2]. Beyond defining static baselines, the carbon gap framework provides a dynamic indicator of how far land-use systems have diverged from their natural equilibrium and how much carbon can be regained through restoration or improved management[3–6]. By integrating current soil organic carbon (SOC) stocks and their potential under reference conditions, this approach reveals spatial patterns of carbon resilience across biomes, linking soil processes to land-use history and management intensity[7,8].

In the context of Brazil's Nationally Determined Contributions (NDCs, 2024) under the Paris Agreement, which target a 59–67% reduction in greenhouse-gas emissions by 2035 and carbon neutrality by 2050[9], the carbon gap concept offers a practical basis for climate mitigation and sustainable agriculture. Agriculture, responsible for about 24–35%[10,11] of national emissions, plays a dual role as both a source and a sink of carbon. Assessing the soil carbon gap supports national low-carbon programs such as the ABC⁺ Program (now RenovAgro), informs the design of credible soil-based carbon credits, and

identifies priority regions for sequestration and restoration, serving as a practical basis for evaluating and improving the Agriculture and Land Use Change initiatives[12–14]. As such, it bridges scientific evidence, policy targets, and market mechanisms to promote soil-centered climate action at scale.

In this study, we compiled a comprehensive database of 4290 $SOC_{stocks}$ spanning Brazil's major biomes (Amazon, Caatinga, Cerrado, Atlantic Forest, Pampa, and Pantanal) to establish a consistent baseline of $SOC_{stocks}$ in native and agricultural areas, at depths of 0–10, 0–20, 0–30, and 0–100 cm. A total of 1247 paired observations of native vegetation (NV) and agricultural systems (AGR) were analyzed to quantify the carbon gap and estimate the theoretical soil recarbonization potential for the 0–30 cm layer, where most management-induced changes occur. We assessed the variability of $SOC_{stocks}$ across biomes and the effects of land-use conversion (NV to AGR), taking into account the main soil and climate classes, as well as the age of agricultural systems. Furthermore, we assessed the influence of different management intensity levels by classifying management systems into three hierarchical levels encompassing five categories: (i) single-crop systems (annual crops, perennials, or grassland); (ii) two-crop systems

[1]Department of Soil Science, Luiz de Queiroz College of Agriculture, University of São Paulo, Av. Pádua Dias, 11, Piracicaba, São Paulo, Brazil. [2]Embrapa Digital Agriculture, Av. Dr. André Tosello, 209, Campinas, São Paulo, Brazil. [3]Graduate Program in Agronomy, State University of Ponta Grossa, Av. General Carlos Cavalcanti, 4748, Ponta Grossa, Paraná, Brazil. [4]Center for Carbon Research in Tropical Agriculture (CCARBON), University of São Paulo, Avenida Pádua Dias, 11, Piracicaba, São Paulo, Brazil. ✉e-mail: cepcerri@usp.br

(rotations or intercropping); and (iii) multi-crop systems (integrated agricultural systems).

## Results and Discussion

### SOC$_{stocks}$ baseline of NV and AGR in Brazilian biomes

The Atlantic Forest (rainforest) presented the highest average values of SOC$_{stocks}$ under NV and AGR in the four layers analyzed (0–10, 0–20, 0–30 and 0–100 cm), whereas the lowest SOC$_{stocks}$ were found in the Pantanal (tropical wetland) and Caatinga (dryland forest) biomes (Fig. 1, Supplementary Tables S1–3, Data 1). In the 0–10 cm layer, the Atlantic Forest's NV SOC$_{stocks}$ were higher ($p < 0.001$) than those of the Caatinga (86%) and Cerrado (savanna) (36%), while its value under AGR surpassed the Pantanal's and Caatinga's by 154% and 62%, respectively. In the subsequent layer, the SOC$_{stocks}$ under NV in the Atlantic Forest were 42–107% greater ($p < 0.001$) than in the Caatinga,

Pantanal, Amazon (rainforest), and Pampa (native grassland), while under AGR, they exceeded these biomes' stocks by 12–188%. The measures in the Atlantic Forest under AGR were also larger (10–108%) than in the aforementioned biomes in the 0–30 cm layer, while under NV, they were greater by 29–116%, except in the Pampa. In the 0–100 cm layer, the Atlantic Forest presented higher SOC$_{stocks}$ than the Amazon and Caatinga, both under NV (79–88%) and AGR (54–63%).

The SOC$_{stocks}$ in the Cerrado were also larger ($p < 0.001$) than those in the Caatinga in the first three layers under NV (34–60%) and in all layers under AGR (47–86%). The biome exhibited greater SOC$_{stocks}$ ($p < 0.001$) than the Pantanal in the first three layers under AGR, with a difference range of 90–157%, and then the Amazon in the last three, with a range of 18–44%. The levels were also higher ($p < 0.001$) than those of the Pampa (17%) in the AGR 0–20 cm layer and of the Amazon in the AGR 0–30 cm (18%).

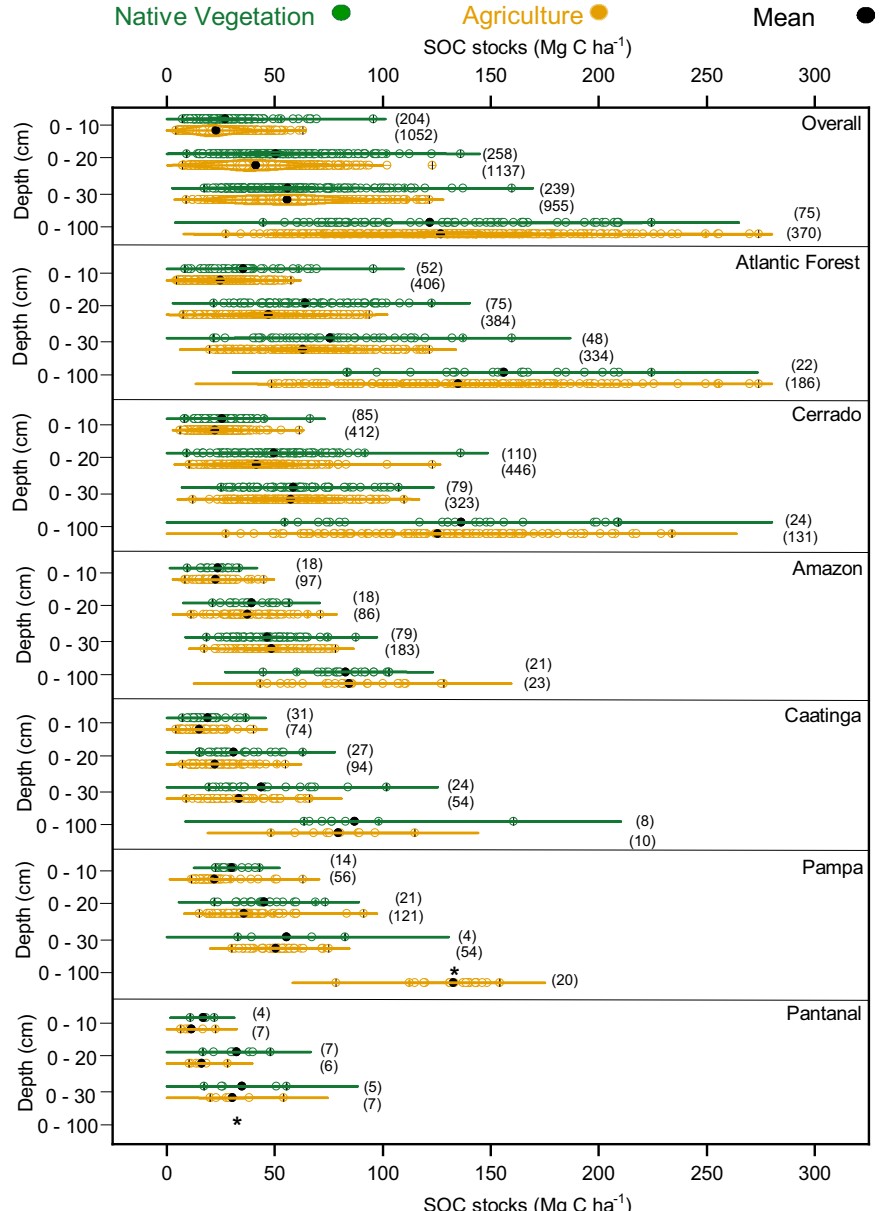

**Fig. 1 | Comparison of SOC$_{stocks}$ under agriculture and native vegetation across Brazil and its biomes.** In the violin plots, dots (black) indicate average (mean) values for SOC$_{stocks}$ under native vegetation - NV (green dots) and agriculture - AGR (orange dots), and the kernel density plot represents the distribution of SOC$_{stocks}$ among land uses overall in Brazil and across its biomes for soil layers 0–10 cm, 0–20 cm, 0–30 cm, and 0–100 cm. The values for each depth layer are independent and should not be interpreted cumulatively. *Data Absence.

$SOC_{stocks}$ in the Pampa and Amazon biomes differed significantly ($p < 0.001$) from those in the Caatinga under agricultural land use, particularly in the first three soil layers (45–68%), and from those in the Pampa under NV in the 0–10 cm layer (59%). The $SOC_{stocks}$ in both biomes also exceeded ($p < 0.001$) those in the Pantanal in the 0–10 and 0–20 cm layers under AGR by 90–131%. Finally, the $SOC_{stocks}$ under AGR were 52% and 62% higher ($p < 0.001$) in the 0–100 cm layer in the Pampa than those in the Amazon and Caatinga, respectively.

The climate is an important factor controlling SOC storage at regional scales, in which precipitation and temperature influence plant growth, the input of organic material, and its rate of decomposition and mineralization[15–17]. We identified a reduction in $SOC_{stocks}$ with increasing temperature, corroborating previous studies[18,19]. Brazil is covered by three climate zones (Köppen's climate classification): Tropical (A), Humid Subtropical (C), and Dry (B), distributed across 81%, 14%, and 5% of the territory, respectively[20]. Our results confirmed higher $SOC_{stocks}$ in more humid and cold regions (i.e., Humid Subtropical>Tropical>Dry) (Supplementary Tables S4-5; Supplementary Fig. S1). The differences ($p < 0.05$) in layers 0–10, 0–20 and 0–30 cm between $SOC_{stocks}$ of the Humid Subtropical and Dry zones were 70–91% (NV) and 58–168% (AGR), while between Tropical and Dry, they were 31–62% (NV) and 50-131% (AGR) (Supplementary Tables S6–9), and between Subtropical and Tropical, they were 17-40% (NV) and 12–16% (AGR). More specifically, $SOC_{stocks}$ under NV and AGR in most soil layers of the biomes occurred in the following climatic order: Tropical rainforest-Af, Temperate (no dry season/warm summer)-Cfb and Temperate (dry winter/warm summer)-Cwb > Temperate (dry winter/hot summer)-Cwa, Temperate (no dry season/hot summer)-Cfa, Tropical Monsoon-Am, Tropical Savannah (summer rain)-Aw, Tropical Savannah (winter rain)-As > Arid (steppe/ hot)-BSh (Supplementary Tables S10,11).

In addition to climatic effects, variations in soil bulk density (BD) further influenced SOC distribution across biomes and land uses (Supplementary Fig. S3). Under NV, BD reflected intrinsic soil-forming processes, including mineralogy, clay content, and pedogenic development. The lowest BD values were observed in Amazon and Atlantic Forest soils (1.05–1.25 Mg m$^{-3}$), whereas higher values occurred in the Pampa and Caatinga (1.25–1.45 Mg m$^{-3}$). After agricultural conversion, BD increased across all biomes, indicating compaction and structural loss. These changes were closely associated with the magnitude of $SOC_{stocks}$ reductions, highlighting the interaction between physical degradation and carbon depletion.

Across all evaluated biomes, $SOC_{stocks}$ under NV and AGR in the first three layers (topsoil: 0–10, 0–20, and 0–30 cm layers) for the five main soil classes decrease in this order: Inceptisol > Oxisol > Ultisol > Entisol > Alfisol (see Supplementary Tables S12 and 13, Supplementary Fig. S2). Under NV, $SOC_{stocks}$ in the first three Inceptisol soil layers were 58–142% higher ($p < 0.05$) than in Alfisol and 44–127% higher than Entisol, while under AGR, the differences were 53–94% and 100–130%, respectively. When comparing the stocks in Inceptisol with the ones in Ultisol, there was less variation, which averaged 47% in the first two layers under NV and 43% in the topsoil under AGR. Finally, Inceptisol $SOC_{stocks}$ were, on average, 16% greater than Oxisol $SOC_{stocks}$ in the 0–10 and 0–30 cm layers.

The $SOC_{stocks}$ of Oxisol also exceeded those of Entisol by 23-83% in the topsoil under NV and by 72-106% under AGR. Relative to Alfisol, Oxisol $SOC_{stocks}$ were 31–67% higher in the topsoil under AGR and, on average, 85% in the 0–20 and 0–30 cm layers under NV. Compared to Ultisol, Oxisol $SOC_{stocks}$ in the topsoil were, on average, 25% higher under AGR and 17-30% in the 0–20 and 0–30 cm layers under NV. Ultisol was 40% higher than Entisol in the 0–30 cm layer under NV and 19-55% higher in the topsoil under AGR.

The highest $SOC_{stocks}$ observed in Inceptisol are due to the formation of histic O or humic A horizons with high accumulation of organic matter, which occur in environments with rugged relief and low temperatures throughout the year[21,22]. The high $SOC_{stocks}$ in Oxisol can be explained by the high stability of aggregates, influenced by the strong association of SOC with Fe and Al sesquioxides. Organo-mineral interaction is the main driver of carbon stabilization in Brazilian tropical soils[23]. The lower $SOC_{stocks}$ observed in Alfisol can be associated with two major drivers: i) lower annual carbon inputs by the Caatinga forest and agriculture in Brazilian drylands, and ii) sandy texture, which reduces carbon protection and stabilization, favoring carbon losses by mineralization and erosion processes[24–27].

Overall, the high proportion of areas with a subtropical climate, the presence of soils that favor carbon accumulation and high altitudes, as identified in our analysis (Supplementary Fig. S4a, b, Supplementary Table S14), contributed to the higher $SOC_{stocks}$ observed under NV and AGR areas within the Atlantic Forest biome, as was also observed in a previous study in this same biome[28]. These characteristics reveal the large mitigation potential in the biome, already demonstrated by adopting best management practices[29–36]. Only in this biome, approximately 20 Mha[37] of pastures are classified into medium and low vegetative vigor, commonly termed "degraded", available for recovery. In contrast, drier climates (semi-arid) and soils highly susceptible to erosion, such as Alfisol and Entisol, led to lower $SOC_{stocks}$ for the Caatinga and Pantanal.

In the Cerrado, the highland tropical climate in the Central Plateau combined with the significant presence of Oxisol contributed to the high levels of $SOC_{stocks}$ in the biome[27]. On the other hand, in the Amazon, despite presenting high rainfall rates, high temperatures reduce the SOC storage potential due to the increased mineralization rate[38]. However, studies of the Amazon have shown that adequate management of integrated agricultural systems[22,39] and improved grassland[40–43] can generate accumulations equal to or greater than NV's.

Despite the extensive database on SOC in native and agricultural areas, substantial gaps remain, particularly in the Pantanal, where only three studies are available and without any data for the 0–100 cm layer. Similarly, the Caatinga, Amazon, and Pampa show limited data coverage at deeper depths, and records of NV in the Pampa are absent. Future research should, therefore, prioritize standardized sampling of deeper soil layers (0–100 cm) and long-term monitoring, especially in underrepresented regions, such as the Pantanal, to enhance the spatial representativeness of national $SOC_{stocks}$ assessments.

## Soil carbon gap between NV and AGR

Considering all biomes, the gaps between $SOC_{stocks}$ under NV and AGR in the 0–10, 0–20, and 0–30 cm layers were respectively −4.4 Mg ha$^{-1}$, confidence interval (CI): [−5.0, −3.8], − 6.9 Mg ha$^{-1}$, CI [−7.7, −6.1], and −5.1 Mg ha$^{-1}$, CI [−6.0, −4.3] (Fig. 2b; Supplementary Table S15 and Data 2). These results revealed that NV-to-AGR conversion resulted in the total loss of 1.4 ± 0.1 Pg C in the 0–30 cm layer (Fig. 2c; Supplementary Table S16.), which is equivalent to an emission of 5.2 Pg CO$_2$eq for the whole country's agricultural area. Although the estimated value indicates a substantial negative effect of land-use change on $SOC_{stocks}$, it also highlights the large carbon sink potential and presents an excellent opportunity to recarbonize agricultural soils in Brazil. Accumulation potential across biomes varied widely, mainly due to the agricultural management practices, but also to the size of the gap between $SOC_{stocks}$ of NV and AGR, to the amount of agricultural land available, and to other variables, including biophysical characteristics, climate, soil class and mineralogy. About 72% of this potential is distributed among the Cerrado (0.53 Pg C) and Atlantic Forest (0.48 Pg C) biomes, which hold the largest agricultural areas in Brazil, respectively 93 Mha and 72 Mha (Fig. 2c, Supplementary Table S16). Approximately 23% of the total potential is in the Amazon (0.19 Pg C) and Caatinga (0.14 Pg C) biomes, which have the third and fourth largest agricultural

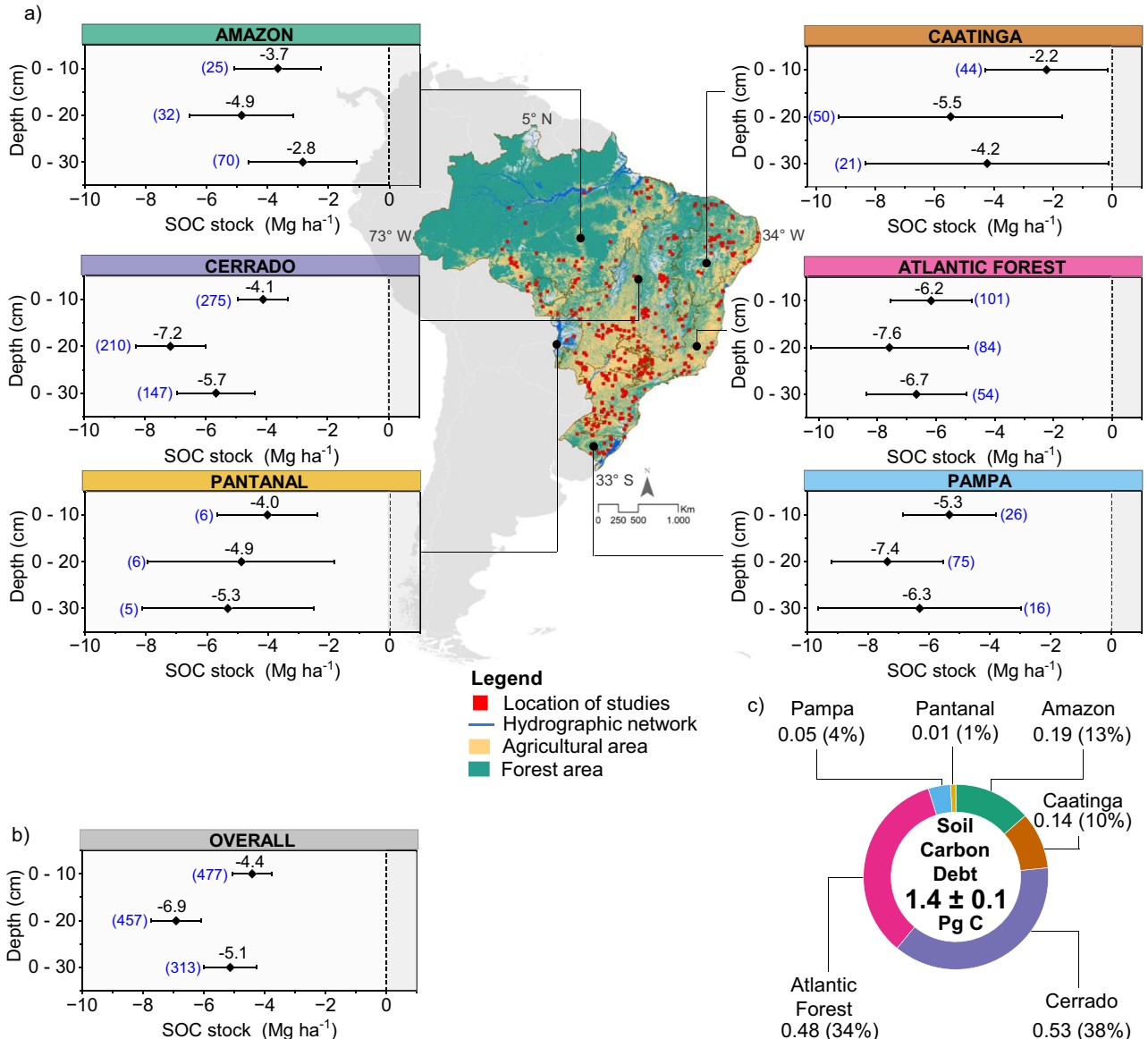

**Fig. 2 | Soil carbon stock changes and sequestration potential across Brazilian biomes. a** Location of studies of SOC_stocks under agriculture (AGR) and native vegetation (NV) and gaps in soil C stocks (Mg ha−1) of the biomes at the depths 0−10, 0−20 and 0−30 cm, obtained from the effect size (mean difference) of land use change (NV to AGR); **b** Overall gaps in SOC_stocks (Mg ha⁻¹); **c** Distribution of soil C sequestration potential (Pg C) across the Brazilian biomes. The bars represent the confidence interval (CI: 95%) of the gap values of soil C stocks and the values next to each mean point () represent the size of the sample evaluated (as presented in Table S1 in supporting information S1). The values for each depth layer are independent and should not be interpreted cumulatively. In Fig. 2a, the free biome boundary data were obtained from IBGE, the data on native vegetation and agricultural areas were obtained from MapBiomas – Coleção 7 of the Annual Land Use and Land Cover Mapping Series (https://brasil.mapbiomas.org/downloads/), and the hydrographic network data were obtained from the ANA metadata catalog (https://metadados.snirh.gov.br/). The data used to map the distribution of the study sites are available in Supplementary Data 1.

areas (67 and 33 Mha, respectively). The lowest potentials were found in the Pampa (0.05 Pg C) and Pantanal (0.01 Pg C) biomes, which comprise only 5% of the total agricultural area.

Gaps in SOC_stocks in biomes varied significantly (36-64% differences between maximum and minimum values). The Atlantic Forest presented the largest gaps in SOC_stocks in the three layers (0−10 cm: -6.2 Mg ha⁻¹, 0−20 cm: -7.6 Mg ha⁻¹, and 0−30 cm: -6.7 Mg ha⁻¹). In contrast, the Amazon and Caatinga exhibited the smallest gaps in SOC_stocks in the three layers, with the lowest value in the 0−10 cm layer observed in the Caatinga (-2.2 Mg ha⁻¹), and in the 0−10 and 0−30 cm layers in the Amazon (-3.7 Mg ha⁻¹ and -2.8 Mg ha⁻¹, respectively). These contrasting patterns suggest that climatic and pedogenic factors strongly mediate the response of SOC_stocks to land-use conversion.

To explore the mechanisms underlying this variability, we assessed the influence of potential moderators (i.e., mean annual temperature, accumulated precipitation, and latitude) on the observed differences between soil layers and biomes. The explanatory power of the moderating variables ranged widely from 1% to 35% of the total heterogeneity across studies (Supplementary Table S17–22). Variability was investigated in more depth through subgroup analyses, considering climate and soil classes, system conversion age, and, at a more detailed level, temperature and precipitation classes in each biome. Based on the average values of the 0–10, 0–20, and 0–30 cm layers, the subtropical climate class presented the greatest reductions, from −11.8% to −22.7% (average = −17.2%), followed by the tropical climate, from −7.5% to −19.4% (average = −13%) (Supplementary

Table S23). The arid climate showed the lowest average percentage loss in $SOC_{stocks}$ (−11.7%) and lacked evidence of a significant effect. This result suggests that NV-to-AGR conversion does not substantially alter carbon dynamics, possibly due to the small initial difference between stocks under the two land uses.

In tropical climate regions, the Atlantic Forest showed the largest losses in $SOC_{stocks}$ (−17.4%), followed by the Caatinga and Cerrado (−12.3%), then the Amazon (−10.5%). Under subtropical climates, similar losses were observed in the Atlantic Forest and Pampa (approximately −17.4%), while the Cerrado showed smaller reduction (-9.8%). In Atlantic Forest areas under the classes Cfa, Cfb, Cwa, and Cwb, which represent approximately 66% of the biome area, losses ranged from -4.4 to -29% and under Aw, present in about 20% of the biome, ranged from -18.1 to -26% (Supplementary Tables S24). In the Cerrado, losses observed under the Aw class, which covers about 75% of the biome, were smaller (-10.4% to -15.2%), while losses under the Aw climate (about 73% of the biome) in the Pantanal exceeded those in the Atlantic Forest and Cerrado, by an average of -22.5%. In the Amazon, losses related to the Am class, which corresponds to about half of the biome, ranged from -7.5% to -13.6%. In the Pampa, Cfa class losses, which encompass almost the entire biome, ranged from -11.8% to -22.3%.

The difference between carbon losses resulting from agricultural conversion under subtropical and tropical or arid climates, ranging from 32% to 47%, is related to the greater sensitivity of soils in cold regions to disturbances. The magnitude of carbon loss is closely linked to initial soil carbon levels. Carbon-rich soils, though naturally stable, undergo greater absolute losses when converted to agriculture due to enhanced organic matter decomposition from soil disturbance and oxygen exposure[11,12]. Consequently, carbon-dense biomes such as the Atlantic Forest and Pampa, covered by subtropical climates, tend to exhibit larger carbon gaps after land-use conversion. In contrast, in tropical and arid climate regions, the effect of conversion is less pronounced, since the difference in $SOC_{stocks}$ is smaller, resulting from faster decomposition and intense carbon turnover, which may have reflected in the smaller gaps recorded in the Caatinga and Amazon.

The analysis of the different soil classes, integrating the three evaluated layers, indicated that average carbon losses decrease with the degree of pedogenetic development: Entisol ≈ Inceptisol (-29% and -28%) > Ultisol (-21%) > Oxisol (-11%), demonstrating the greater susceptibility of young soils to C loss after land-use conversion (Supplementary Table S25). The Atlantic Forest showed the greatest carbon losses in Inceptisol (-33%), followed by the Cerrado (-28%), and the Pampa and Amazon biomes (both with -13%). The percentage loss recorded in the Atlantic Forest Ultisols was almost double (-24%) that observed in the Caatinga (-11%). In Oxisol, the Atlantic Forest showed the greatest loss (-15%), followed by the Cerrado and Amazon with -11% and -9%, respectively. The greatest carbon loss in Entisol was observed in the Pantanal (-25%), followed by the Caatinga (-23%) and Amazon (-20%). The losses observed in the Caatinga's Inceptisols, the Amazon's and Pampa's Ultisols, the Pampa's and Caatinga's Oxisols, and the Cerrado's Entisols did not show statistical significance, possibly due to the influence of factors such as the management system or climatic conditions.

The analysis of the average age of conversion of agricultural systems revealed that the Pampa and Atlantic Forest have the oldest systems (31 years), followed by the Cerrado (26 years), Caatinga (19 years), Amazon (18 years), and Pantanal (16 years). When evaluating the effect of conversion age (0–15, 16–30, and >30 years) on $SOC_{stocks}$, it was found that carbon losses occur across biomes, being more intense in the first 15 years after conversion (Supplementary Table S26). The Amazon showed significant losses of approximately -11% in the first 15 years after conversion, which were reduced to -7% between 16 and 30 years, without statistical significance. In systems older than 30 years, losses remained around -9%, indicating low recovery of $SOC_{stocks}$ over

time. In the Atlantic Forest, SOC stocks decreased by approximately -15% in systems up to 30 years old and by -19% in older systems. Although the difference is modest, this pattern suggests that SOC losses may persist over time following conversion to AGR, highlighting the biome's vulnerability to land use change. The Caatinga recorded a significant reduction of -17% in the first 15 years after conversion, decreasing in subsequent periods, ranging from -15% to -6%, but without statistical significance. In the Cerrado, carbon loss during the first 15 years was lower (-9.4%) compared to the Caatinga (-17%), Atlantic Forest (-15%), and Amazon (-11%), increasing to -12% in the subsequent periods.

The Pampa biome was the only one to show an increase in $SOC_{stocks}$ (2%) in systems up to 15 years old, although this increase was not statistically significant. This result may reflect transitions in management (tillage to no-tillage or to integrated agricultural systems) or the presence of residual C from the original vegetation. Finally, the Pantanal biome recorded the greatest C loss, averaging 24% up to 30 years, reflecting the high sensitivity of the biome's Entisol soils to the conversion process.

The proportion of studies in certain age groups across biomes, soil and climate classes may be reflected in the estimated C gap. The largest gaps observed in the Atlantic Forest and Pampa biomes may be related to the higher proportion of studies in systems over 30 years old (48% and 53%, respectively), age groups that showed the most pronounced losses compared to systems in other age groups. In contrast, the smaller gaps in the Amazon and Caatinga may reflect the predominance (~ 80%) of data from systems up to 30 years old, which exhibited a dual behavior: while some results were close to zero, the 16−30 year group showed negative but non-significant results, indicating high variability and low C losses.

The larger C gap in the 0−20 layer may be related to the disturbance caused by mechanical tillage[44,45], which accelerates the decomposition of soil organic matter (aggregates disruption, increased microbial activity, and aeration) and, consequently, reduces the SOC contents in the soil surface[46]. The high variability between $SOC_{stocks}$ gaps in the biomes, mainly in the 0−10 and 0−30 cm layers, can be explained by the different edaphoclimatic conditions, soil management, and land use history in the ecoregions[18,23,47].

Our estimate of SOC debt (1.40 ± 0.1 Pg C) in the 0−30 cm layer in agricultural lands represents the aggregate total for Brazil compiled from all biomes and is consistent with a global study which also estimated national SOC loss for Brazil (1.6 Pg C[4]), although without biome-scale resolution. Thus, our estimate provides a more detailed understanding than large-scale studies, allowing us to identify regional patterns, biome responses, and management factors associated with SOC changes. It is important to emphasize that our estimate represents theoretical potential, based on a hypothetical scenario assuming the restoration of original $SOC_{stocks}$ through soil recarbonization, and should therefore be interpreted with caution. Another aspect that should be considered and acknowledged as a methodological limitation is the use of SOC levels under NV as a fixed reference in estimates. Although this approach is widely adopted, in contexts involving degraded native ecosystems or highly improved agricultural lands, this assumption may not be fully applicable, potentially resulting in an underestimation or overestimation of the SOC gap and accumulation potential.

Despite these limitations, our results provide the best quantitative benchmark for the achievable $SOC_{stocks}$ to this date. It allows us to estimate an order of magnitude of the opportunity: by recarbonizing approximately *1/3* of the estimated potential, we could achieve the 59−67% emissions reduction target for 2035 (1.51–1.71 Pg $CO_2$eq, base year: 2005) in Brazil's NDC[2]. In practice, the target would be achieved by harnessing approximately 40−45% of the recarbonization potential accumulated in the Atlantic Forest and Cerrado (3.8 Pg $CO_2$eq).

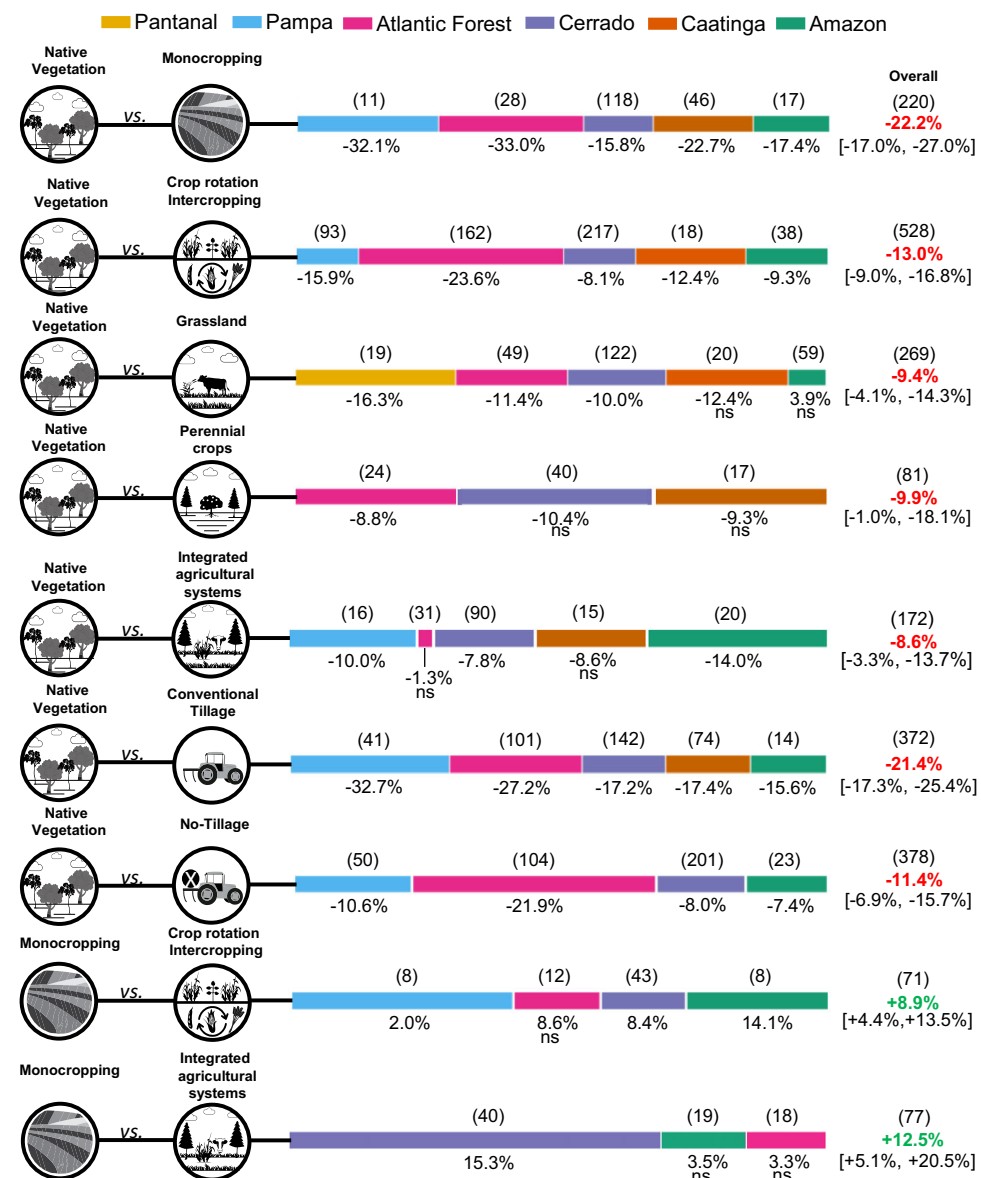

**Fig. 3 | Percent change in SOC$_{stocks}$ resulting from land-use and management transitions in Brazilian biomes.** The bars represent the mean effect sizes expressed as percent change ($\pm$ 95% CI of the response ratio-lnRR), estimated using a random-effects model. Values in parentheses indicate the number of paired observations used to calculate changes in SOC$_{stocks}$ among land-use and management systems, whereas values in brackets represent the confidence intervals. The notation ns indicates non-significant statistical differences at the 95% confidence interval. In the overall panel, negative effects are shown in red and positive effects in green.

## Impact of agricultural system management on SOC$_{stocks}$

In a more detailed analysis of the SOC$_{stocks}$ gap in Brazilian agricultural lands, we assessed the impacts of converting NV to different agricultural production systems on SOC$_{stocks}$ at combined depths of 0–10, 0–20, and 0–30 cm (Fig. 3, Supplementary Table S3 and Supplementary Data 3–11). Considering all biomes, NV-to-monocropping conversion showed the highest percentage of change, -22.2%, CI: [-27.0, -17.0], followed by NV-to-crop rotation/intercropping, -13.0%, [-19.6, -16.0] (Fig. 3). The Atlantic Forest and Pampas showed the highest relative reductions in SOC under monoculture (−33.0% and −32.1%, respectively), while the Atlantic Forest also had the highest loss under crop rotation/intercropping (−23.6%) (Table S27). In contrast, the Cerrado and Amazon recorded the smallest reductions under both systems, ranging from −15.8% to −8.1% under monoculture and from −17.4% to −9.3% under rotation/intercropping. The Caatinga exhibited intermediate relative changes, with

SOC$_{stocks}$ losses of -22.7% under monocropping and -12.4% under crop rotation and intercropping.

Grasslands presented, on average, a SOC$_{stocks}$ change of -9.4%, CI [-14.3, -4.1] SOC$_{stocks}$ in comparison with NV. The greatest loss was observed in the Pantanal, with -16.3%, and the smallest in the Cerrado (-10.0%) and Atlantic Forest (-11.4%) biomes. The Caatinga and Amazon did not present a significant difference ($p > 0.05$).

Perennial crops showed similar mean SOC$_{stocks}$ variations (9.9%, CI [-18.1, -1.0]) to those of grasslands (-9.4%). The Atlantic Forest had the smallest difference (-8.8%), while the Cerrado and Caatinga recorded larger ones (-10.4% and -9.3%), without statistical significance, possibly due to high variability between areas or partial stabilization of SOC$_{stocks}$ in systems with low soil disturbance and continuous input of organic residues.

Integrated agricultural systems had the lowest relative change (-8.6%, IC [-10.7, -4.7]), showing average losses lower than those

observed in monoculture, crop rotation/intercropping, and grassland, respectively 61%, 34%, and 9%. Compared to NV, integrated agricultural systems showed the largest difference in the Amazon (-14.8%), and the smallest significant difference in the Cerrado (-7.8%), followed by the Pampa (-10%). The differences in the Atlantic Forest (-1.2%) and Caatinga (-8.6%) were not statistically significant (Fig. 3, Supplementary Table S27). These results revealed the benefits of agricultural intensification, understood within the framework of sustainable intensification, in which yields increase without adverse environmental impacts or the conversion of additional non-agricultural land[48], mainly by biodiversification, to preserve/restore $SOC_{stocks}$.

Our results also showed the positive effect of intensification through the comparison between both crop rotation/intercropping and integrated agricultural systems with monocropping. The crop rotation and intercropping system presented an average gain of +8.9%, IC [ +13.5, +4.4], with the Pampa increasing the most ( + 22%), followed by the Amazon ( + 14.0%), and Cerrado, with the lowest value (+8.4%), while the Atlantic Forest's increase ( + 8.6%) had no statistical significance. Under integrated agricultural systems, the average gain related to monocropping was +12.5%, IC [ + 20.5, +5.1], i.e., 40% higher than the crop rotation and intercropping value, with the Cerrado showing a gain of 15.3%. There was no significant difference ($p > 0.05$) between the Atlantic Forest under both systems and the Amazon under integrated agricultural systems.

We also compared $SOC_{stocks}$ losses from NV to no-tillage and conventional tillage systems. Our results confirmed that losses under no-tillage (-11.4%, CI [-15.7, -6.9]) were 47% lower than under conventional tillage (-21.4%, CI [-25.4, -17.3). The Pampa and Atlantic Forest recorded the most significant losses under the conventional system, respectively -32.7% and -27.2%. The loss in the Pampa was 88–110% greater than in the Cerrado (-17.2%), Caatinga (-17.4%), and Amazon (-15.6%) biomes, while in the Atlantic Forest it exceeded these biomes by 58–74%. Under the no-tillage system, the Atlantic Forest recorded the greatest loss of $SOC_{stocks}$ (-21.9%), followed by the Pampa (-10.6%), while the Cerrado and Amazon presented the smallest losses of -8.0% and -7.4%, respectively. We also evaluated the effect of the no-tillage system on different soil depths to identify possible variations associated with the stratification induced by this management. The results (Supplementary Fig. S5) indicated a small variation between the original mean value (-11.4%) and those obtained in the stratified analysis (0–10 cm = -14.0%; 0–20 cm and 0–30 cm ≈ -10.0%), with a more pronounced difference in the 0–10 cm layer and slightly smaller ones in the subsequent layers.

Soil carbon losses induced by land-use change are well-documented in the literature. Our results, ranging from -9.4% to -26.3%, are comparable with previous studies in the tropical[49] and global scale[50]. However, these values can be even more variable (-11.4% to -95.6%), depending on context, when evaluated in a global perspective[51–54]. Our study showed that the magnitude of $SOC_{stocks}$ losses in different agricultural systems decreases with the increase in their degree of intensification and diversification, confirming the benefits induced by crop rotation, intercropping[55–61], and integrated agricultural systems[62–68], reported in different regions around the world.

For NV-to-grassland conversion (-4.1 to -14.3%), our results corroborate studies of tropical zones[49] and of global scope (-9.8 to -14.4%)[69], although other studies reported a positive effect (11.5 to 8.2%)[51,52]. Best management practices (defoliation regime, plant species manipulation, biodiversity, fertilization, improved grazing, grass-legume intercropping, and irrigation)[70–73] and continuous grassland maintenance[74], in addition to the effect of local soil and climate conditions[1,75], may explain the positive response.

The absence of significant differences observed in the Caatinga and Amazon biomes denotes the variation in the maintenance status of Brazilian grassland areas. The greater losses observed in the other biomes reflect the current condition of Brazilian grasslands, with ca. 60% of the area found at intermediate and severe levels of degradation[40]. In this sense, grassland recovery is an important technology in Brazil, as it can boost $SOC_{stocks}$ by 14–23%[76]. It has been adopted by national climate mitigation initiatives (ABC+ Program[77] and National Program for the Conversion of Degraded Pastures into Sustainable Agricultural and Forestry Production[78]), which project the recovery of 40 Mha as a target[77].

Evidence from this study highlights the role of integrated agricultural systems as effective strategies for maintaining and rebuilding $SOC_{stocks}$, with the smallest differences observed in the Pampa and Cerrado and statistically non-significant differences in the Atlantic Forest and Caatinga. Compared with monocropping, using the technology in the Cerrado showed great potential to increase $SOC_{stocks}$, with values close to those reported in a global study ( + 17% to +23%)[1]. The ABC+ Program's portfolio of solutions includes plans to expand these systems by 4 Mha, with the potential to mitigate up to 34 million Mg $CO_2$eq.

The benefits of no-tillage over conventional tillage revealed in this study were consistent with estimates previously reported for Brazil (6-26%)[79,80]. Our analysis demonstrates the potential of no-tillage as an alternative to replace conventional tillage based on differences from NV. The area with no-tillage in Brazil increased significantly between 2006-2017 (17.9 to 33.0 Mha)[81], (4–16% in $SOC_{stocks}$), replacing conventional tillage[82]. The ABC+ Program set a goal to expand the system by 12.5 Mha by 2030, expecting to mitigate 12 million Mg $CO_2$eq.

Although our results reveal the positive effects of agricultural intensification, these benefits depend on local conditions such as climate, soil type, and duration of management. When supported by conservation-oriented practices, such as no-tillage, crop rotation diversification, cover cropping, residue retention, and the use of organic amendments, intensification can enhance SOC inputs, improve aggregation and biological activity, and foster long-term carbon stabilization[2,83], while promoting positive feedbacks to crop yield and climate resilience[84,85].

It is important to note that the $SOC_{stocks}$ gap estimates presented here represent a technical potential based on an idealized reference of NV. In reality, native ecosystems may be partially degraded, and improved agricultural systems can already recover part of their $SOC_{stocks}$. Thus, our values should be interpreted as indicative of the upper limit of carbon restoration potential, rather than as directly attainable field targets.

In conclusion, this national-scale study provided quantitative evidence of the soil carbon debt of 1.40 ± 0.1 Pg (0–30 cm) induced by historic land-use change of NV to AGR across the Brazilian biomes. In addition, it revealed the positive effect of sustainable management intensification (i.e., crop rotation and intercropping, no-tillage, and integrated agricultural systems) on increasing $SOC_{stocks}$, which helps close this carbon gap over time. Finally, these results support the formulation of regionally adapted intervention strategies, which could strengthen Brazil's climate policies and help mitigate global climate change.

## Methods

We performed a first-order meta-analysis to examine the impacts of land-use change from NV to agricultural (AGR) habitats on $SOC_{stocks}$ of the six Brazilian biomes. A meta-analysis was performed using data from 372 studies (Cerrado:141, Atlantic Forest:129, Amazon:38, Pampa:34, Caatinga:28 and Pantanal:3) from the last three decades (Supplementary Fig. S6 and Discussion). Our analysis examined the gap between $SOC_{stocks}$ under NV and AGR, the soil C gap, which we consider our technical potential for C accumulation. We also assessed the magnitude of the positive and negative effects of different agricultural production and management systems on $SOC_{stocks}$.

## Systematic literature search

The systematic review was conducted by searching the Web of Science, Scopus, and Scielo databases (see Supplementary Fig. S6 and S7 for details) for relevant studies using data collected in the Brazilian territory until June 2024, without language restriction. We employed the following combinations of keywords: "soil carbon stock* OR soil carbon OR soil organic* AND CU= (Brazil)".

Studies were selected based on the following eligibility criteria: i) having presented $SOC_{stocks}$ data or data that allowed the calculation of $SOC_{stocks}$ (soil organic matter and soil BD); ii) studies carried out in Brazil, with data on geographic coordinates or the name of the municipality or biome. Using these criteria, we found 4290 data points, 776 of which pertained to primary NV and 3514 to agriculture (Supplementary Data 1).

## Data collection

We collected the following data from text, tables, or figures in the selected studies (using *WebPlotDigitizer* https://automeris.io/WebPlotDigitizer):land-use classes (NV or AGR), soil (U.S. soil taxonomy) and climate classes (in the absence of information, we used the data available in ref. 20), agricultural systems (monocropping, crop-rotation and intercropping, grasslands, perennial crops, and integrated agricultural systems), $SOC_{stocks}$ in the soil layers (0–10, 0–20, 0–30 and 0–100 cm), numbers of observations, and standard deviations. These were recorded in a data framework. $SOC_{stocks}$ estimates for the different soil depth intervals represent independent measurements derived from the available data and should not be interpreted as cumulative profiles. Because complete soil profiles were not consistently reported across studies, the values for each depth layer reflect discrete sampling intervals rather than integrated depth totals. $SOC_{stocks}$ data were standardized by converting the values from "kg m$^{-2}$," "g m$^{-2}$," "t ha$^{-1}$," "kg ha$^{-1}$," or "percentage" to "Mg ha$^{-1}$". The SOC concentration data (g kg$^{-1}$) were converted to $SOC_{stock}$ (Mg C ha$^{-1}$) from the soil BD and specific depth (cm) using the following equation[52,86]:

$$SOC\ stock\ (MgC\ ha^{-1}) = \frac{SOC\ x\ BD\ x\ layer\ thickness}{10} \quad (1)$$

## Classification and comparison of management systems

The "agriculture" land-use class was subdivided into agricultural production and management systems, described as follows. "Monocropping" involves a single cultivated agricultural crop species. Grassland, eucalyptus, pine, and other perennial fruit crops were redistributed to other categories. Crop rotation (planned alternation of crop species over time) and intercropping (changes between growing seasons) form a category of agricultural systems where two or more crops are grown sequentially in the same field. Such systems that included grassland in the cycle were transferred to the "Grassland" category, which comprises cultivated and native grassland. The latter accounts for 3% of the total category data set. "Perennial crops" are planted forest species such as Eucalyptus spp. and Pinus spp., which represent 65% of the data set, as well as other perennial fruit species. "Integrated agricultural systems" encompass crop–livestock integration, crop–forest integration, forest–livestock integration, and crop–livestock–forest integration. "Conventional tillage" is soil preparation using techniques that promote soil disturbance, such as plowing, harrowing, and leveling, whereas "No-tillage" systems do not disturb the soil with tillage: seeds are sowed directly into straw or the previous crop's residues. The study evaluated the effects of monocropping, crop rotation and intercropping, grasslands, perennial crops, and integrated agricultural systems again NV baselines. Additionally, conventional tillage and no-tillage management systems were compared to both NV and to monocropping agricultural production systems.

## Meta-analysis

From the primary database, we selected a subset based on the following eligibility criteria for the meta-analysis: presenting paired data on $SOC_{stocks}$ with their respective standard deviations (SD) and number of observations. For studies lacking SD values for $SOC_{stocks}$ measurements, missing SDs were imputed using the approach based on median coefficients of variation[87,88], following these steps: (1) calculated coefficients of variation (CV) from available SD and $SOC_{stocks}$ values; (2) estimated the median CV separately for each dataset (treatment and control); and (3) computed missing SDs by multiplying the median CV by the reported $SOC_{stocks}$ for that group. To maintain natural variability and minimize bias, imputation was stratified by biome, land-use class, and soil layer. We tested the effect of different imputation assumptions by conducting a sensitivity analysis[89], which showed only marginal change in means and 95% confidence intervals with different imputed SD values, indicating our findings were robust (Supplementary Table S28 and Fig. S8). The final database contains data from 2004-2024 (considering the 20-year IPCC steady-state period), presented in 1,247 pairwise comparisons (NV vs. AGR) and grouped by layer depth (0–10 cm, 0–20 cm, and 0–30 cm) and by biome.

We selected the weighted mean difference (MD) as a dependent variable of the meta-analysis to measure the effect size of the NV-to-AGR transition. The MD and Natural log of response ratio (lnRR) were calculated to measure the effect of different agricultural and management systems on $SOC_{stocks}$ using the following equations:

$$MD = \mu_{e} - \mu_{c} \quad (2)$$

$$lnRR = \ln\frac{\mu_{e}}{\mu_{c}} = \ln\ln(\mu_{e}) - \ln\ln(\mu_{c}) \quad (3)$$

where $\mu_{c}$ and $\mu_{e}$ correspond to the average $SOC_{stocks}$ before (control) and after (treatment) the change in land-use/management practices[46], respectively.

To quantify the effect size of the NV-to-AGR transition, we compared estimates obtained from two models: a random-effects model and a linear mixed-effects model[90]. Both types of models were fitted using restricted maximum likelihood estimation. For the mixed-effects model, we evaluated the inclusion of nine moderator variables, both categorical and continuous, categorized by temporal dynamics, climate and soil classes, geography, and soil management systems, nested by biome and soil layer. We used p-values (considered significant when <0.05, indicating a direct effect) and $R^2$ (the proportion of explained heterogeneity) as primary criteria for including or excluding moderators. As secondary criteria, we examined indicators related to heterogeneity diagnostics: $\tau^2$ (between-study variance), $I^2$ (proportion of variance due to heterogeneity), and $H^2$ (heterogeneity ratio), which quantify the magnitude of remaining heterogeneity[90]. In practical terms, a significant increase ($p < 0.05$) in explained heterogeneity ($R^2$), along with reductions in heterogeneity indicators compared to the random-effects model, indicates that the variable contributes to explaining variability. Based on these statistical indicators and on the main objective of our meta-analysis, which was to estimate the technical potential for carbon accumulation in agricultural soils across different biomes from the effect of NV-to-AGR transition on $SOC_{stocks}$, our model ultimately included only three moderator variables: latitude (°), mean annual temperature (°C), and cumulative annual precipitation (mm).

To enhance interpretability of coefficients, these continuous moderators were centered and scaled prior to their inclusion in the models. Additionally, a subgroup meta-analysis was performed to

complement the global meta-analysis and investigate the effects of new moderating variables, which included climate and soil classes (described in Supplementary Tables S4 and S12), as well as the age of the agricultural system, considering the year of conversion and the date of soil sampling. The age classes were 15, 30, and over 30 years. This complementary approach allowed for a more granular investigation into specific contextual factors that might not have been fully captured by the global variables from the primary model, providing a nuanced understanding of $SOC_{stocks}$ dynamics across different biomes and management contexts.

For the step of the meta-analysis procedure where we assessed the effect of different agricultural and management systems on $SOC_{stocks}$, the data of the three soil layers (0–10, 0–20, and 0–30 cm) were analyzed jointly using the random-effects model (Supplementary Data 3-11). To address the non-independence of overlapping soil depth measurements within the same study site, we employed multilevel hierarchical models[91], using the rma.mv function. This approach explicitly models the nested structure of our data (soil depths nested within studies) through random effects (random = ~ 1 | study_id/depth_raw). This methodology allows us to: (1) utilize all available soil carbon data, rather than discarding 60% of observations by selecting only one depth per study; (2) properly account for the correlation between overlapping soil layers from the same location; (3) avoid pseudoreplication and the consequent underestimation of variance; and (4) provide more conservative and accurate statistical inferences. This approach is considered the gold standard for meta-analyses dealing with non-independent observations and has been widely recommended in the ecological meta-analysis literature. In the case of the comparison between no-tillage and NV, in addition to the general analysis considering the three soil layers (Fig. 3), the effect of each layer was also evaluated individually (Supplementary Fig. S5) to detect differences associated with carbon stratification. The results indicated that the 0–10 cm layer presented greater $SOC_{stocks}$ depletion (14.6%) than the 0–20 and 0–30 cm layers, which presented an average loss of 11.6%.

All effect size calculation procedures were performed in the R software, using the MD (raw mean difference, e.g[92].) and ROM (log-transformed ratio of means[87],) functions of the R package "Metafor"[90], considering that the effect sizes varied randomly between the comparisons. The effects measured to assess changes in $SOC_{stocks}$ of agricultural and management systems (ln RR) were transformed and expressed as a percentage change using the following equation[93]:

$$\triangle SOC \ stock(\%) = \exp[\ln(RR) - 1] \times 100 \qquad (4)$$

The overall effect sizes and 95% confidence intervals (CI) were also calculated using the "Metafor" package. Statistically significant changes in $SOC_{stocks}$ were considered valid when the 95% CI value did not overlap zero. The CIs of $SOC_{stocks}$ effect sizes for NV-to-AGR transition were shown in forest plots (Fig. 2), where negative values indicated C losses. For agricultural and management systems, although results were not presented in forest plots (Fig. 3; Supplementary Table S27), we also interpreted negative percentages as C losses and positive ones as C gains.

### Sensitivity analyses and publication biases

We assessed the potential for publication bias (Supplementary Fig. S9 and S10) using the funnel plot and Egger's test[94]. The test assesses the existence of publication bias through regression analysis using the effect size and its precision (standard error). A significant intercept (indexed by b), i.e., with $p < 0.05$, suggests publication bias[95]. To assess the sensitivity of the results to publication bias, Rosenthal's fail-safe number[96] was applied, which estimates the number of hypothetical studies with a null effect required to nullify the significance of the combined effect (Supplementary Table S17-20). The trim-and-fill

procedure was also used to adjust the mean effect value and thus take possible missing studies into account. The findings from the combination of the three methods confirm that the results of the meta-analyses by biome are, for the most part, statistically robust and little affected by publication bias. The extremely high Rosenthal's fail-safe N values and the absence of trim-and-fill adjustments reinforce the reliability of the estimates. The Cerrado showed the greatest sensitivity to potential bias in the 0–10 cm layer, but, even in this case, the overall effect remained significant and stable. It is worth noting that methods such as Egger's test and the funnel plot have their own limitations, especially in contexts with high heterogeneity of the observed effects[97], which applies to our study.

### Calculation of potential $SOC_{stocks}$ sequestration

The technical potential of $SOC_{stocks}$ sequestration overall and for each biome was calculated by the following equation:

$$SOC \ stock \ seq. = Gap_{biomes}(Mg \ C \ ha^{-1}) \times AGR_{area} \ (Mha) \qquad (5)$$

where $SOC_{stocks}$ seq. represents the technical potential for C sequestration (Pg C). $Gap_{biomes}$ represents the gap between AGR (treatment) - NV (control) in Mg C ha$^{-1}$ and $AGR_{area}$, the available agricultural area in the biomes in Mha.

### Statistical analysis

The comparison between the $SOC_{stocks}$ (dependent variable) under NV and AGR across climate and soil classes, as well as across biomes, was made employing the ANOVA test, the Tukey's post hoc test for normal distribution, and the nonparametric Kruskal-Wallis (univariate) when the normality condition was not met, all of them for a statistical significance level of 5%. The analyses were performed using the "rstatix" package in the R software (R Core Team, 2021)[98].

## Data availability

All data supporting the findings of this study are available within the paper and its Supplementary Information files.

## Code availability

All code used for data processing, statistical analyses and figure generation has been deposited in a publicly accessible GitHub–Zenodo repository under the https://doi.org/10.5281/zenodo.17842918.

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

## Acknowledgements

J.V. acknowledges support from the São Paulo Research Foundation (FAPESP, grant #2023/09533-3). J.D. acknowledges support from the São Paulo Research Foundation (FAPESP, grant #2022/08629-4). M.R.C. acknowledges support from the National Council for Scientific and Technological Development (CNPq, grant #302249/2025–7). This research was supported by: (i) the Center for Carbon Research in Tropical Agriculture (CCARBON/USP), sponsored by FAPESP (grant #2021/10573-4); (ii) the Research Center for Greenhouse Gas Innovation (RCGI/USP), sponsored by FAPESP (grant #2020/15230-5) and Shell Brasil; and (iii) Bayer's PROCarbono project. The authors thank M.Sc. Julie Camolesi da Silva and Professor Fabricio Batistin Zanatta for their assistance with statistical analyses, and Marina Pinho Oncken for language review services.

## Author contributions

J.V., M.C., J.D., and C.C. designed the study. J.V. and J.D. collected data. J.V. performed the meta-analysis. J.V. wrote the article, with significant contributions provided by M.C., J.D., D.G., L.B., and C.C. All the authors contributed to the discussions and paper revision.

## Competing interests

The authors declare no competing interests.
