## [Peer Review file · Nature Communications]

Soil carbon debt from land use change in Brazil

Corresponding Author: Professor Carlos Eduardo Cerri

Version 0:

Reviewer comments:

Reviewer #1

(Remarks to the Author)

Revealing the carbon debt in certain regions is crucial for achieving carbon net-zero thus mitigating the changing climate. Using literature reporting carbon stocks in Brazil, Villela et al. analyzed the difference in carbon stock between natural vegetation and farmland, which aimed to address the potential of farmland in Brazil in global C market. The topic was interesting while the findings, the methods, and the writing were not fulfilled with the readers, which may hinder its publishing in Nature Communications, at least at its current status.

1. The authors emphasized their findings as the carbon debt of farmland in Brazil was 1.41 Pg C in Abstract, which may make reader misunderstand the role of farmland in global C market. Actually, the debt is the difference between farmland and natural vegetation, while it was impossible to transform all farmland to natural vegetation as human beings needed sufficient farmland to support their livings. In addition, the findings that the reported farmland carbon debt in Brazil was similar to previous works, which provided no novel information.
2. The methods in drawing the carbon debt were too arbitrary. Using the average carbon stocks of farmland and natural vegetation after integrating the farmland area of each biome, the authors calculated their differences thus the carbon debt. However, the carbon stocks varied substantially even within the same biome. Only employing the average values would miss much information and may give biased results.
3. It was strange to see the results of SOC stock baselines after analyzing the carbon debt and their affecting factors. Such results were employed to calculate the carbon debt and played little role in documenting the main findings of this manuscript. If the authors intended to describe those results, I suggested you put them to the first part of Results.
4. I found the carbon debt in 0-30cm layer were lower thus that in 0-10cm layer, which means that farmland may have a higher carbon stock in 20-30cm layer. This result was strange, which may be related to the uneven distributions of measurements and should be fully addressed.
5. I did not see Figure 1 and the units of x-axis in Figure 4.

Reviewer #2

(Remarks to the Author)

This manuscript presents a timely and policy-relevant assessment of carbon debt in Brazilian agricultural soils, using meta-analytic techniques to quantify soil organic carbon (SOC) losses and recarbonization potential across different biomes and land-use systems. The scope and ambition of the study are commendable, and the dataset is both extensive and well-curated. The Introduction is well-written, and the Results and Discussion are generally strong and clearly presented. However, as the methodological approach forms a central pillar of the article, my review has focused primarily on this aspect. Several key methodological elements require clarification and enhancement to ensure the robustness and interpretability of the findings. Addressing these issues may necessitate revisions to the Results and Discussion sections, especially where findings are sensitive to the assumptions or modeling approaches used. While the study makes a valuable contribution to understanding SOC dynamics and mitigation potential in Brazilian agriculture, I recommend that the manuscript undergo major revisions before publication. These improvements will strengthen the analytical rigor and reinforce the manuscript's policy relevance, ensuring it meets the high standards expected for publication in Nature.

1. Treatment of Missing Variance Data:

The authors impute a fixed 7% standard deviation for studies lacking variance estimates, based on internal averages. While defensible, this approach introduces uncertainty into effect size calculations. The analysis would be considerably strengthened by including a sensitivity analysis to assess the robustness of the findings under alternative SD assumptions

or, preferably, by applying stratified imputation based on biome or land use system. This would reduce potential bias and enhance confidence in the meta-analytic outputs.

2. Absence of Temporal Dynamics:

A significant limitation is the absence of “time since land-use change” or management intervention as a moderating variable. SOC is temporally dynamic, particularly following deforestation or adoption of conservation practices. Including temporal information would allow for differentiation between short-term losses and longer-term stabilization or gains, which is particularly important for informing mitigation policy and MRV systems.

3. Limited Integration of Contextual Variables:

While the study provides stratified descriptive analysis by biome, soil class, and climate zone, these important contextual factors are not statistically integrated into the meta-analytic models. Incorporating them as moderators within the random-effects framework would enable more nuanced interpretation of SOC responses and reduce unexplained heterogeneity.

4. Lack of Heterogeneity Diagnostics:

Although the use of a random-effects model is appropriate, the manuscript does not report standard heterogeneity metrics such as I^2 or τ^2 . Including these statistics is essential to evaluate between-study variability and to justify the use of the chosen model structure. It would also help readers understand the distribution and consistency of effect sizes across the dataset.

5. Assumptions Behind the SOC Baseline and Potential:

The use of native vegetation SOC levels as a fixed baseline for estimating sequestration potential is a common practice but warrants greater scrutiny. In certain cases—such as degraded native systems or highly improved agricultural systems—this assumption may not hold. The authors should acknowledge this limitation and qualify the interpretation of the SOC “gap” and technical potential accordingly, particularly in the discussion and conclusions.

6. Other general comments

Villela et al. used meta-analysis of an unedited soil organic carbon stock (SOCstock) baseline in Brazil (4,261 data points) and estimated the potential of recarbonizing the agricultural lands of the six Brazilian biomes (Amazon, Caatinga, Cerrado, Atlantic Forest, Pampa, and Pantanal). They selected 1,306 pairwise comparisons of native vegetation (NV) vs. agricultural systems (AGR) and estimated the carbon sequestration (accumulation?) potential for the 0–30 cm layer. They also assessed the effect of three different agricultural system intensification levels of monoculture; crop succession and rotation; and integrated agricultural systems compared to NV in Brazil. Results were presented as a comparative evaluation between SOCstocks of NV and AGR for topsoil and the 0–100 cm layer. They considered the “soil carbon gap” obtained by the difference between SOCstocks from NV and AGR. Results from this meta-analysis indicated a carbon debt of $1.41 \pm 0.3 \text{ Pg C}$ (0–30 cm) for agriculture and a positive effect of agricultural intensification on increasing SOCstocks. No-tillage compared to conventional soil management systems with NV decreased SOCstock losses. This is an important study, and it should be published after revision. This study reveals the size of the opportunity for carbon farming in Brazil to contribute as a major player in the emerging global C market.

Specific comments

- Figure 1 is not shown in the manuscript.
- Figure 2c: carbon sequestration potential or carbon accumulation potential?
- Figure 3: The extreme left circle (some circles do not have a label)
- Figure 4: X-axis units?
- Since the authors estimated SOCstocks using soil bulk density (BD) values, the differences in average BD values of six different biomes may be either presented as a Figure (Supplementary Information?) or the main observations described in the Results section.
- The Discussion section highlights the importance of climate and soil type, and it may be strengthened regarding the role of BD and its influence on SOCstock losses.

Reviewer #3

(Remarks to the Author)

The manuscript titled “Carbon debt in cropland soils of Brazil” presents a well-structured and timely contribution addressing one of the most pressing global environmental challenges—carbon sequestration and emissions associated with land use, particularly in the Brazilian context. Given Brazil’s pivotal role as both an agricultural powerhouse and a major player in global carbon dynamics, the findings are of high international relevance. The authors successfully identify carbon gaps across key Brazilian biomes, highlighting substantial opportunities for carbon restoration through land management strategies.

Major Comments:

Scientific Relevance and Novelty:

The study offers valuable insights by quantifying carbon debt across diverse biomes, reinforcing the potential for land management to mitigate carbon losses. The use of a comprehensive and current database strengthens the findings;

however, the lack of data for certain biomes—especially the Pantanal—represents a notable limitation. A more robust discussion on this gap would enhance the study's completeness.

Figures and Presentation:

The figures are generally well-designed and effectively support the narrative. However, there appears to be a discrepancy in figure numbering—specifically, there is no Figure 1. This should be corrected either by renumbering or including the missing figure. Additionally, several textual statements are not clearly linked to figures, and in some instances, the depth associated with reported results is missing. Clarifying these points will improve readability and data traceability.

Language and Style:

Although the manuscript contains minor grammatical and stylistic errors, these do not detract significantly from the overall clarity. A careful revision by a native English speaker or a professional editing service is recommended to enhance fluency and polish.

Methodological Rigor:

The methodology appears sound; however, it is described too briefly in the current version. More detailed information in the Materials and Methods section is needed to allow for full reproducibility. For example, further clarification is required on how different land uses and management intensities were categorized and analyzed.

Title Accuracy:

The current title may be somewhat misleading, as the analysis extends beyond cropland to include grasslands and native soils. A more precise title would better reflect the manuscript's scope.

Conceptual Clarity:

The abstract and introduction would benefit from improved clarity. Several key topics—such as land use intensification, soil types, and climate analysis—are introduced abruptly or not at all in the early sections. For instance, three land uses are introduced initially, but five are later discussed, including grasslands, which are not clearly integrated into the conceptual framework. These inconsistencies weaken the alignment between the study's objectives, methods, and results.

Management Trade-Offs:

One critical issue that warrants deeper discussion is the dual role of agricultural intensification. While such practices can enhance soil carbon stocks under certain conditions, they can also lead to negative soil quality outcomes—such as compaction—particularly due to increased machinery traffic in high-frequency crop rotations. Additionally, it should be emphasized that intensification strategies may only be sustainable on soils with high agricultural suitability; applying them to fragile or marginal lands could exacerbate degradation.

Minor Comments:

Ensure that all statements in the Results section are directly supported by corresponding figures and data.

Clarify the depth of soil sampling where it is not currently specified.

Detailed comments are noted in the Manuscript pdf.

Version 1:

Reviewer comments:

Reviewer #1

(Remarks to the Author)

I have carefully evaluated the revised manuscript and am pleased to see that the authors have fully and convincingly addressed all of my previous comments. The revisions substantially improve the clarity, rigor, and completeness of the work. I find the manuscript to be scientifically sound, clearly presented, and suitable for publication.

Only very minor issues remain, primarily related to reference formatting. A few entries contain small typographical errors (e.g., references No. 21, 37, and 77). I recommend a final thorough check of the reference list for consistency (author names, journal titles, page ranges, and punctuation) before acceptance.

Overall, I am satisfied with the revisions and recommend the manuscript for publication in Nature Communications after minor editorial corrections.

Reviewer #2

(Remarks to the Author)

After reviewing the document carefully, I do not have any additional comments or suggestions at this moment. The content is clear and reflects the discussions accurately.

Reviewer #3

(Remarks to the Author)

Dear Authors,

I believe that this second version has greatly improved from the previous one. In my opinion all the suggestions made were solved. In the attach I highligh some minor mistakes or small comments to take into consideration. All the best. The Reviewer.

- Reviewer's comments in *blue*
- Replies in black

REVIEWER COMMENTS

Reviewer #1 (Remarks to the Author):

Revealing the carbon debt in certain regions is crucial for achieving carbon net-zero thus mitigating the changing climate. Using literature reporting carbon stocks in Brazil, Villela et al. analyzed the difference in carbon stock between natural vegetation and farmland, which aimed to address the potential of farmland in Brazil in global C market. The topic was interesting while the findings, the methods, and the writing were not fulfilled with the readers, which may hinder its publishing in Nature Communications, at least at its current status.

1. The authors emphasized their findings as the carbon debt of farmland in Brazil was 1.41 Pg C in Abstract, which may make reader misunderstand the role of farmland in global C market. Actually, the debt is the difference between farmland and natural vegetation, while it was impossible to transform all farmland to natural vegetation as human beings needed sufficient farmland to support their livings. In addition, the findings that the reported farmland carbon debt in Brazil was similar to previous works, which provided no novel information.

R: We appreciate this valuable comment and fully agree that the term carbon debt refers to the difference in SOC stocks between agricultural lands and native vegetation, and does not imply that farmland could or should be reverted entirely to natural ecosystems. To avoid any potential misunderstanding, we have revised the Abstract and Discussion to clarify that the value of 1.40 Pg C represents a theoretical carbon debt, that is, the estimated gap relative to native conditions, which serves as a reference to quantify the potential for soil carbon recovery through recarbonization practices, rather than a scenario of full land-use reversion.

Specifically, we have:

1. Reframed the introduction to clearly emphasize the conceptual basis and global relevance of the soil carbon gap approach, linking it to the Paris Agreement, Brazil's Nationally Determined Contributions (NDCs), and the role of agriculture in carbon neutrality (lines 28–47).

2. Clarified the objectives and novelty of our study, highlighting that our analysis provides the first biome-scale quantification of the soil carbon gap across six major Brazilian ecosystems, integrating 4,290 site-level observations into a harmonized framework. This allows us to identify regional patterns, biome-specific responses, and

management-related drivers of SOC change, offering a more detailed and context-specific understanding than previous large-scale syntheses. (lines 48–60).

3. Improved the meta-analysis performance including new moderators (e.g., mean annual temperature, accumulated precipitation, and latitude). In addition, we also evaluated the effect of the bulk density (BD) variable on the distribution of SOC stocks under different land uses among biomes (lines 201–214).

4. Results and Discussion sections to better communicate the significance of regional variability, the mechanisms behind SOC stock changes, and the implications for carbon recovery and market-based mitigation strategies (lines 171–469).

5. Enhanced the writing and structure throughout the text to improve flow, precision, and accessibility to a broader audience of Nature Communications.

We believe that these revisions significantly strengthen the manuscript and address the reviewer's concern regarding scientific rigor and communication.

2. The methods in drawing the carbon debt were too arbitrary. Using the average carbon stocks of farmland and natural vegetation after integrating the farmland area of each biome, the authors calculated their differences thus the carbon debt. However, the carbon stocks varied substantially even within the same biome. Only employing the average values would miss much information and may give biased results.

R: We appreciate your consideration of the methodology used to calculate the debt. We agree that SOC stocks can vary significantly within biomes. This variability is influenced by differences in climate, soil properties, and the agricultural profile of each region. Therefore, in addition to estimating the mean value, we included confidence intervals to highlight the variability observed across studies in each biome.

As an alternative to providing greater detail in the debt estimates between biomes, we tested a regional approach with assessments divided by latitude intervals. However, after a general assessment, we found exceptions. Only the Cerrado biome, which had the highest data availability, allowed for this analysis. For other regions, we could not apply this approach due to a limited number of comparative pairs ($n < 10$). Comparing standard and regional approaches for the Cerrado showed similar precision (Figure 1.a). The average values by region (-6.6 and -4.9%) remained within the confidence interval (IC) estimated by the standard approach (-6.9 to -4.4%).

To strengthen the robustness of our results, we included new moderators in this revision round of the meta-analysis, which enhanced our ability to account for the spatial

and temporal variability of SOC stocks across different biomes. Therefore, we consider the methodology applied in this study to be valid and capable of adequately capturing the variability observed among regions within each biome. Nonetheless, we acknowledge that regional-scale approaches can provide additional granularity on carbon debt, offering stronger technical support for decision-making processes.

Figure 1. (a) Comparison between carbon gap values obtained from the general and regional (latitude-based) approaches in the Cerrado biome; (b) Map showing the spatial distribution of carbon gaps across regions within the Cerrado biome.

3. It was strange to see the results of SOC stock baselines after analyzing the carbon debt and their affecting factors. Such results were employed to calculate the carbon debt and played little role in documenting the main findings of this manuscript. If the authors intended to describe those results, I suggested you put them to the first part of Results.

R: Thank you for this valuable comment and for highlighting the need for a clearer logical flow between sections. We fully agree that the SOC stock baseline results are foundational to the subsequent analysis of carbon debt and should therefore be presented earlier in the manuscript. In response, we have reorganized the Results section, moving the baseline analysis to the beginning of the text. This restructuring improves the narrative coherence, allowing readers to first understand the spatial distribution and magnitude of SOC stocks before exploring the derived carbon debt patterns and their controlling factors. The baseline results were plotted in Figure 1.

4. I found the carbon debt in 0-30cm layer were lower thus that in 0-10cm layer, which means that farmland may have a higher carbon stock in 20-30cm layer. This result was strange, which may be related to the uneven distributions of measurements and should be fully addressed.

R: Thank you for this valuable methodological observation. We agree that the apparent inconsistency between carbon debt values across depth intervals may reflect the uneven distribution of samples and the incomplete coverage of soil profiles in our dataset. To clarify this point, we added a statement in the *Methods* section (lines 493–495) explaining that the carbon gap values should be interpreted independently by depth interval, as full soil profiles including all three depths (0–10, 0–20, and 0–30 cm) were not consistently available. In addition, a note was added to Figure 2 (“The values for each depth layer are independent and should not be interpreted cumulatively”) to clarify this methodological aspect.

5. I did not see Figure 1 and the units of x-axis in Figure 4.

R: Thank you for noting this issue with figure numbering and axis labeling. The figure has been revised to clearly display the SOC stock units (Mg ha^{-1}) on both the upper and lower x-axes. Following the reorganization of the Results section, the previous “Figure 4” is now presented as “Figure 1”. These adjustments improve consistency between figure references and enhance overall readability.

Reviewer #2 (Remarks to the Author):

This manuscript presents a timely and policy-relevant assessment of carbon debt in Brazilian agricultural soils, using meta-analytic techniques to quantify soil organic carbon (SOC) losses and recarbonization potential across different biomes and land-use systems. The scope and ambition of the study are commendable, and the dataset is both extensive and well-curated. The Introduction is well-written, and the Results and Discussion are generally strong and clearly presented. However, as the methodological approach forms a central pillar of the article, my review has focused primarily on this aspect. Several key methodological elements require clarification and enhancement to ensure the robustness and interpretability of the findings. Addressing these issues may necessitate revisions to the Results and Discussion sections, especially where findings are sensitive to the assumptions or modeling approaches used. While the study makes a valuable contribution to understanding SOC dynamics and mitigation potential in Brazilian agriculture, I recommend that the manuscript undergo major revisions before publication. These improvements will strengthen the analytical rigor and reinforce the manuscript's policy relevance, ensuring it meets the high standards expected for publication in *Nature*.

R: We sincerely thank the reviewer for this thorough and constructive evaluation, as well as for recognizing the scientific and policy relevance of our study. We fully agree that methodological transparency and robustness are critical to the strength of our conclusions. Accordingly, we have undertaken a comprehensive revision of the *Methods*, *Results*, and *Discussion* sections to clarify our analytical framework and strengthen the interpretation of findings.

Specifically, we:

1. **Expanded the methodological description** to better detail data harmonization, inclusion criteria, and the weighting structure used in the meta-analytic models (lines 464–646).
2. **Clarified assumptions and sensitivity analyses**, explicitly discussing the implications of data imbalance across biomes, depth intervals, and land-use categories (lines 612–629; Table S28).
3. **Added explanatory notes** to ensure that depth-specific results are interpreted independently, avoiding misinterpretation of SOC differences as cumulative profiles (lines 493–495).
4. **Revised the Results and Discussion** to explicitly link methodological assumptions to key findings and to qualify statements where data heterogeneity could influence interpretation (lines 171–469).

These revisions enhance the analytical rigor and transparency of our approach, addressing the reviewer's concerns while preserving the clarity and policy relevance of the manuscript.

We are confident that the revised version now meets the methodological standards expected for publication in *Nature Communications*.

1. Treatment of Missing Variance Data:

The authors impute a fixed 7% standard deviation for studies lacking variance estimates, based on internal averages. While defensible, this approach introduces uncertainty into effect size calculations. The analysis would be considerably strengthened by including a sensitivity analysis to assess the robustness of the findings under alternative SD assumptions or, preferably, by applying stratified imputation based on biome or land use system. This would reduce potential bias and enhance confidence in the meta-analytic outputs.

R: We thank the reviewer for this valuable methodological comment. We agree that imputing a generic 7% standard deviation on the mean carbon stocks can introduce uncertainty into effect size calculations. Following this recommendation, we revised our analytical procedure to implement a stratified imputation of missing SD values by biome, land-use system, and soil layer. For each group (treatment and control), we calculated the median coefficient of variation (CV) based on studies that reported both means and SDs, following the procedures described by Lajeunesse (2013) and Koricheva et al. (2013). Missing SDs were then estimated by multiplying the median CV by the reported mean SOC stock for each group.

Thus, missing standard deviation values were filled in by multiplying the median coefficient of variation by the mean of the control and treatment groups. When no grassland data with complete statistics were available, we used the median CVs from agricultural systems within the same layer. For the Pantanal biome, where SD data were largely unavailable, CVs from the geographically and pedogenetically similar Cerrado biome were adopted.

To assess the robustness of our findings under alternative SD assumptions, we performed a sensitivity analysis (Melikov et al., 2023). Based on the set of coefficients of variation obtained for each of the classes described in the stratification, we calculated the median, first, and third quartiles (Table 1). We then replaced the missing standard deviations from the original dataset with standard deviations using the median, first, and third quartiles and subjected the resulting datasets to meta-analysis, with the results presented in Figure 2. The results indicate robustness of the analysis, since for most of the dataset, the mean and 95% confidence intervals underwent only marginal changes with the variation in the imputed standard deviations. Thus, we consider that the new standard

deviation imputation method adopted reduces potential biases associated with uniform imputation, allowing greater robustness and reliability to the results of the meta-analysis.

Table 1. Median coefficients of variation were applied when standard deviations were unavailable, and the first (Q1) and third quartiles (Q3) were used for sensitivity analyses.

Biomes	Soil layer (cm)	Land use (CV%)								
		Crop			Grassland			Native vegetation		
		Q1	Median	Q3	Q1	Median	Q3	Q1	Median	Q3
Amazon	-									
	0 - 10	5.8	9.1	20.6	4.3	7.2	8.2	5.2	6.5	12.4
	0 - 20	3.3	4.2	5.6	2.2	3.2	4.2	5.6	6.6	11.7
Caatinga	0 - 30	3.4	4.9	7.6	5.1	6.4	9.6	3.2	4.8	10.5
	0 - 10	6.0	9.5	11.6	10.5	17.3	24.7	5.0	13.3	16.4
	0 - 20	4.1	5.9	8.5	5.8	7.0	9.6	5.0	6.3	9.4
Cerrado	0 - 30	3.8	4.1	5.2	3.4	3.9	4.3	4.2	5.0	5.9
	0 - 10	3.0	5.1	9.6	2.4	5.2	11.9	6.0	8.9	15.5
	0 - 20	2.5	4.8	7.1	1.6	3.7	5.1	3.3	7.0	8.1
Atlantic Forest	0 - 30	2.0	2.9	4.7	3.1	4.4	5.5	3.4	4.5	7.8
	0 - 10	4.2	6.1	10.5	5.1	5.8	12.5	3.5	4.5	5.4
	0 - 20	2.7	3.6	6.2	6.4	7.8	9.3	2.2	2.8	4.0
Pampa	0 - 30	1.9	2.6	5.1	5.2	5.2	14.6	1.8	3.2	5.1
	0 - 10	2.4	6.0	6.5	*	*	*	***	***	***
	0 - 20	0.7	2.8	6.4	*	*	*	4.1	6.3	8.5
Pantanal	0 - 30	1.0	1.3	1.6	*	*	*	4.5	4.8	5.0
	0 - 10	-	-	-	**	**	**	**	**	**
	0 - 20	-	-	-	**	**	**	**	**	**
	0 - 30	-	-	-	**	**	**	**	**	**

* Median CV from agriculture used when mean and SD were not reported in grassland

** Median Cerrado CVs applied for lack of Pantanal SD

*** In the absence of references for a specific soil layer, we adopted the SD values reported for the 0–20 cm layer.

Figure 2. Sensitivity analysis of SOC stock gaps meta-analysis to imputing missing standard deviations.

Actions: We replaced the sentence that described the methodology originally used in the study “In studies that did not present SD data, we adopted a percentage value of the mean, as suggested by 78, which was defined based on an average SD value of studies in our dataset that contained this information, resulting in a percentage of 7%.”, by the sentence that describes the updated methodological procedure “For studies lacking standard deviation (SD) values for SOC stock measurements, we imputed missing SDs using the approach proposed by Lajeunesse [86] and Koricheva [87], following these steps: (1) calculated coefficients of variation (CV) from available SD and SOC stock values; (2) estimated the median CV separately for each dataset (treatment or control); and (3) computed missing SDs by multiplying the median CV by the reported SOC stock for that group. To maintain natural variability and minimize bias, imputation was stratified by biome, land-use class, and soil layer. We tested the effect of different imputation assumptions by

*conducting a sensitivity analysis (Melikov et al., 2023 [88]). The results showed that the means and 95% confidence intervals changed only marginally with different imputed SD values, indicating our findings were robust (see **Table S28** and **Fig. S7**)."*

We removed reference [78] (Shi, L., Feng, W., Xu, J. & Kuzyakov, Y. Agroforestry systems: Meta-analysis of soil carbon stocks, sequestration processes, and future potentials. *Land Degrad. Dev.* 29, 3886–3897 (2018).) and added the following references:

86. Lajeunesse, M. J. Recovering missing or partial data from studies: a survey of conversions and imputations for meta-analysis. In *Handbook of Meta-analysis in Ecology and Evolution* (eds Koricheva, J., Gurevitch, J. & Mengersen, K.) 195–206 (Princeton Univ. Press, Princeton, 2013).

87. Koricheva, J., Gurevitch, J. & Mengersen, K. *Handbook of Meta-analysis in Ecology and Evolution* (Princeton Univ. Press, Princeton, 2013).

88. Melikov, C. H., Bukoski, J. J., Cook-Patton, S. C., Ban, H., Chen, J. L. & Potts, M. D. Quantifying the effect size of management actions on aboveground carbon stocks in forest plantations. *Curr. Forestry Rep.* 9, 131-148 (2023).

To provide further methodological clarity, we included Table S7 and Figure S28 in the Supplementary Material, illustrating the procedure for imputing standard deviations and conducting the sensitivity analysis.

2. Absence of Temporal Dynamics:

A significant limitation is the absence of “time since land-use change” or management intervention as a moderating variable. SOC is temporally dynamic, particularly following deforestation or adoption of conservation practices. Including temporal information would allow for differentiation between short-term losses and longer-term stabilization or gains, which is particularly important for informing mitigation policy and MRV systems.

R: We thank the reviewer for this important observation regarding the temporal dynamics of SOC change. We fully agree that incorporating “time since land-use change” and “management duration” as moderators could provide valuable insight into short- and long-term carbon dynamics. Following this recommendation, we tested the inclusion of two temporal variables in the meta-analysis: time since land-use conversion (Conv_Time) and time under management (Time_Span) (Table 1, pp. 31-32, at this document).

The effect of including variables in the model was assessed based on statistical indicators used as exclusion criteria. The primary criteria considered were: p-value (significant when <0.05 , indicating a direct effect) and R^2 (%), which expresses the proportion of heterogeneity explained. Secondary criteria analyzed were τ^2 (variance between studies), I^2 (proportion of variance attributed to heterogeneity), H^2 (heterogeneity ratio), and QE (test of residual heterogeneity), which allow measuring the magnitude of the

remaining heterogeneity. In practical terms, a significant reduction in these indicators compared to the null model indicates that the inclusion of the variable contributes to explaining the variability.

It can be observed that the inclusion of variables related to temporal dynamics improved the model's performance only in some biomes and specific layers. Based on R^2 values, the biomes that performed best were: Pampa (0-30 cm = 37% and 0-10 cm = 11.4%), Amazon (0-10 cm = 32.8% and 0-30 cm = 15.9%), and Caatinga (0-30 cm = 17.2% and 0-10 cm = 7.9%). A reduction in indicators related to heterogeneity (τ^2 , I^2 , H^2 , and QE) was also observed. To a lesser extent, a subtle improvement was observed in the 0–10 cm layers of the Caatinga and 0–20 cm layers of the Atlantic Forest. Although the R^2 results for the Pantanal biome indicate model improvement, we recognize that the analysis should not be considered due to the small number of studies in the biome, which could result in inconsistencies in the results. The Cerrado was the only biome in which the inclusion of these variables did not contribute to improved model performance.

Although we observed positive effects of temporal dynamics variables in some biomes (Pampa, Amazon, and Caatinga), we chose not to include them in the final model.

This decision was based on the absence of an explanatory effect in the biomes with the largest number of observations (Cerrado and Atlantic Forest), which provide greater statistical robustness. Therefore, we consider that the selected set of environmental moderators offers greater consistency and parsimony to the model.

Nevertheless, to address the reviewer's suggestion, we further explored these temporal effects through a subgroup analysis (biome \times soil layer), in which agricultural systems were grouped by years since land-use conversion (15, 30, and >30 years). This complementary analysis, described in the *Supplementary Material*, helps illustrate the temporal trends in SOC change even though they were not retained as primary moderators in the final meta-analytic model.

3.Limited Integration of Contextual Variables:

While the study provides stratified descriptive analysis by biome, soil class, and climate zone, these important contextual factors are not statistically integrated into the meta-analytic models. Incorporating them as moderators within the random-effects framework would enable more nuanced interpretation of SOC responses and reduce unexplained heterogeneity.

R: Thank you for your valuable consideration regarding the inclusion of contextual moderators in our analysis. Following your suggestion, we have included the contextual variables available in our database, organized into five categories, as listed below:

- (i) **Temporal dynamics** (time since land use change; time since management intervention);
- (ii) **Climate** (Köppen climate classes; average annual temperature; accumulated annual precipitation);
- (iii) **Soil** (soil class according to the North American classification);
- (iv) **Geography/spatial** (latitude; longitude);
- (v) **Soil management systems** (no-till, conventional tillage, pasture, perennials, forestry, and integrated agricultural systems).

The analysis aimed to identify the most sensitive variables with statistically significant effects. Overall, the results indicated low explanatory power, in addition to estimated intercepts outside a plausible range, which led, in most cases, to non-significant effects (confidence intervals crossing zero). The results, including estimates and heterogeneity diagnosis from the comparative analysis between the random and mixed-effects models with the inclusion of the aforementioned moderators, are available in Appendix I (at this document).

Although the initial set of variables did not explain much of the observed variability, we selected mean annual temperature, accumulated annual precipitation, and latitude to compose the comparative analysis between the mixed-effects model and the random-effects model. The inclusion of these variables is justified by their superior relative performance, albeit with limited explanatory power, and by their well-established physical relationship with soil carbon dynamics (reference variables).

The results of the comparative analysis (Heterogeneity analysis and Bias and robustness analyses) are shown in Supplementary Tables S17-22 in the article. The heterogeneity metrics and the bias and robustness tests indicate that,

in almost all biomes, I^2 values were very high (>90%), confirming strong heterogeneity between studies. The assessment of the explanatory gain, based on the proportion of heterogeneity explained by the moderators (R^2 %), showed better results in Atlantic Forest (4–35%) and the Amazon (17–18%). In the Pampa, the values ranged from 12–19%, while in the Cerrado they ranged from 3–9.2%. In the Caatinga, the moderators had no explanatory effect and, in some cases, worsened the fit (negative R^2 values). The Pantanal, on the other hand, exhibited moderate values (14–24%), but the low number of observations limits the statistical reliability of these results.

Although including moderators provided some explanatory power in certain biomes, the overall effect was limited. It is worth noting that the estimates from both models (random-effects and mixed-effects) showed little variation. Therefore, we present both results (Supplementary Table) and use the estimates from the mixed-effects model to calculate the carbon gap in the biomes.

4.Lack of Heterogeneity Diagnostics:

Although the use of a random-effects model is appropriate, the manuscript does not report standard heterogeneity metrics such as I^2 or τ^2 . Including these statistics is essential to evaluate between-study variability and to justify the use of the chosen model structure. It would also help readers understand the distribution and consistency of effect sizes across the dataset.

R: We appreciate the reviewer’s insightful comment and fully agree that reporting heterogeneity metrics is essential to assess between-study variability, justify model selection, and clarify the consistency of effect sizes across datasets. In response, we have now included a detailed heterogeneity diagnosis in Supplementary Tables S17-22 in the article, presenting I^2 , τ^2 , H^2 , and R^2 values by biome and soil layer for both random-effects and mixed-effects models.

These statistics confirm that variability across studies warranted the use of random-effects modeling, while also supporting the inclusion of moderators in the mixed-effects structure. We believe this addition enhances methodological transparency and allows readers to better interpret the robustness and heterogeneity of our meta-analytic results.

5.Assumptions Behind the SOC Baseline and Potential:

The use of native vegetation SOC levels as a fixed baseline for estimating sequestration potential is a common practice but warrants greater scrutiny. In certain cases—such as degraded native systems or highly improved agricultural

systems—this assumption may not hold. The authors should acknowledge this limitation and qualify the interpretation of the SOC “gap” and technical potential accordingly, particularly in the discussion and conclusions.

R: We thank the reviewer for this thoughtful and important observation. We agree that using native vegetation SOC levels as a fixed baseline may not fully account for site-specific variability—particularly in cases where reference ecosystems are degraded or agricultural systems have undergone significant improvement.

We chose this criterion because native vegetation is the most commonly adopted reference in meta-analyses of soil carbon sequestration^{1–3} and is also used in IPCC greenhouse gas inventory methodologies (Tier 1, 2, and 3) in the AFOLU sector (IPCC, 2019; IPCC, 2006). This approach provides a consistent and comparable baseline across regions and land use systems, enabling a more robust synthesis of global patterns.

Nonetheless, we acknowledge that in certain situations this assumption could lead to over- or underestimation of the SOC gap, depending on the ecological integrity of native sites or the management intensity of agricultural systems. In response to this valuable suggestion, we have added clarifying statements to the *Discussion (lines 292–305)* and *Conclusions (lines 422–426)*, explicitly noting that the reported SOC gaps represent a theoretical or technical potential for carbon accumulation, reflecting an idealized baseline rather than a universally achievable target under field conditions.

We believe this revision provides a more balanced and transparent interpretation of the results while maintaining methodological consistency with international standards.

References

1. **Sanderman, J., Hengl, T. & Fiske, G. J.** Soil carbon debt of 12,000 years of human land use. *Proc. Natl Acad. Sci. USA* **114**, 9575–9580 (2017).
2. **Chai, H. et al.** Drivers of soil organic carbon recovery under forest restoration: a global meta-analysis. *Carbon Research* **1**, 10 (2021).
3. **Wu, L. et al.** Response of soil organic carbon stocks and soil microbial biomass carbon to natural grassland conversion: a global meta-analysis. *Sci. Total Environ.* **852**, 158318 (2022).
4. 1. IPCC (2006). 2006 IPCC Guidelines for National Greenhouse Gas Inventories – Volume 4: Agriculture, Forestry and Other Land Use. Intergovernmental Panel on Climate Change (IPCC), Geneva, Switzerland. Disponível em: ipcc-nggip.iges.or.jp

5. 2. IPCC (2019). 2019 Refinement to the 2006 IPCC Guidelines for National Greenhouse Gas Inventories – Volume 4: Agriculture, Forestry and Other Land Use. Intergovernmental Panel on Climate Change (IPCC), Geneva, Switzerland. Disponível em: https://www.ipcc.ch/report/2019-refinement-to-the-2006-ipcc-guidelines-for-national-greenhouse-gas-inventories/?utm_source=chatgpt.com

6. Other general comments

Villela et al. used meta-analysis of an unedited soil organic carbon stock (SOCstock) baseline in Brazil (4,261 data points) and estimated the potential of recarbonizing the agricultural lands of the six Brazilian biomes (Amazon, Caatinga, Cerrado, Atlantic Forest, Pampa, and Pantanal). They selected 1,306 pairwise comparisons of native vegetation (NV) vs. agricultural systems (AGR) and estimated the carbon sequestration (accumulation?) potential for the 0 – 30 cm layer. They also assessed the effect of three different agricultural system intensification levels of monoculture; crop succession and rotation; and integrated agricultural systems compared to NV in Brazil. Results were presented as a comparative evaluation between SOCstocks of NV and AGR for topsoil and the 0–100 cm layer. They considered the "soil carbon gap" obtained by the difference between SOCstocks from NV and AGR. Results from this meta-analysis indicated a carbon debt of 1.41 ± 0.3 Pg C (0 – 30 cm) for agriculture and a positive effect of agricultural intensification on increasing SOCstocks. No-tillage compared to conventional soil management systems with NV decreased SOCstock losses. This is an important study, and it should be published after revision. This study reveals the size of the opportunity for carbon farming in Brazil to contribute as a major player in the emerging global C market.

R: We sincerely thank the reviewer for this encouraging and thoughtful evaluation. We are pleased that the reviewer recognizes the scientific and policy relevance of our study and its contribution to understanding soil carbon dynamics and recarbonization potential in Brazilian agricultural systems.

Following the reviewers' valuable feedback, we implemented several major methodological and structural improvements to further strengthen the robustness and transparency of our analysis. These include:

- (i) a revised imputation procedure for missing variance data based on stratified coefficients of variation,
- (ii) inclusion of heterogeneity diagnostics (I^2 , τ^2 , H^2 and R^2) by biome and soil layer,
- (iii) testing of temporal moderators related to land-use conversion and management time, and
- (iv) refined interpretation of SOC gap values as representing a technical or theoretical potential for carbon accumulation.

We believe these revisions substantially enhance the analytical rigor and the interpretability of the manuscript, while maintaining its alignment with global carbon accounting frameworks and the policy goals of Brazil's agricultural sector within the context of carbon farming and climate neutrality.

Specific comments

- Figure 1 is not shown in the manuscript/ Figure 4: X-axis units?

R: We thank the reviewer for this helpful observation. The figure numbering and layout have been revised following the reorganization of the *Results* section. The figure originally referred to as “Figure 4” is now presented as Figure 1.

We have also updated the figure to include the SOC stock unit (Mg ha^{-1}) on both the upper and lower x-axes to ensure clarity and consistency with the rest of the manuscript. The revised version has been carefully checked to confirm that all figures are properly referenced and sequentially numbered throughout the text.

- Figure 2c: carbon sequestration potential or carbon accumulation potential?

R: We appreciate the reviewer's attention to terminology consistency. Following this comment, we replaced the term “sequestration” with “accumulation” in the caption of Figure 2c to ensure conceptual precision and alignment with the manuscript's definitions.

The updated caption now reads:

“(c) Distribution of soil C accumulation potential (Pg C) for the Brazilian biomes.”

We also reviewed the text to confirm that the same terminology (“accumulation potential”) is used consistently throughout the manuscript and supplementary materials.

- Figure 3: The extreme left circle (some circles do not have a label)

R: Thank you for your comments regarding the missing labels in the figure. Figure 3 has been updated to include all previously missing labels.

- Since the authors estimated \$\text{SOC}_{\text{stocks}}\$ using soil bulk density (BD) values, the differences in average BD values of six different biomes may be either presented as a Figure (Supplementary Information?) or the main observations described in the Results section.

R: We thank the reviewer for this constructive suggestion. Following this recommendation, we now include a detailed analysis of bulk density (BD) variability across the six tropical biomes. A new supplementary figure (Figure S3)

presents BD distributions under both native vegetation and agricultural land uses. In addition, we expanded the Results section (lines 110–118) to highlight the main patterns observed.

We first analyzed BD for soils under native vegetation, which represents natural reference conditions free from anthropogenic disturbance. Differences among these natural soils reflect intrinsic pedogenic factors such as parent material, clay content, and degree of weathering. Based on our dataset, lower BD values were found in Amazon and Atlantic Forest soils (1.05–1.25 Mg m⁻³), while higher BD values occurred in Pampa and caatinga (1.25–1.45 Mg m⁻³).

When comparing agricultural soils, BD values were consistently higher across all biomes, approximately 1.20–1.35 Mg m⁻³ in the Amazon, 1.25–1.40 Mg m⁻³ in the Atlantic Forest, 1.40–1.55 Mg m⁻³ in the Caatinga, 1.35–1.50 Mg m⁻³ in the Cerrado, and 1.45–1.55 Mg m⁻³ in the Pampa. These increases (0.1–0.2 Mg m⁻³ on average) reflect compaction and organic matter loss due to agricultural management.

We added the sentence below to the results section:

“In addition to climatic effects, variations in soil bulk density (BD) further influenced SOC distribution across biomes and land uses (Supplementary Fig. S3). Under NV, BD reflected intrinsic soil-forming processes, including mineralogy, clay content, and pedogenic development. The lowest BD values were observed in Amazon and Atlantic Forest soils (1.05–1.25 Mg m⁻³), whereas higher values occurred in the Pampa and Caatinga (1.25–1.45 Mg m⁻³). After agricultural conversion, BD increased across all biomes, indicating compaction and structural loss. These changes were closely associated with the magnitude of SOC_{stocks} reductions, highlighting the interaction between physical degradation and carbon depletion.”

• The Discussion section highlights the importance of climate and soil type, and it may be strengthened regarding the role of BD and its influence on SOC_{stock} losses.

R: We appreciate the reviewer’s insightful observation. We have expanded the Results (lines 110–118) to address the influence of BD on SOC_{stock} changes, emphasizing both intrinsic soil characteristics and land-use–induced alterations.

We now highlight that BD under native vegetation represents a natural equilibrium between organic matter input, aggregation, and mineral composition. Literature values support this range: Oxisols (1.2–1.4 Mg m⁻³), Ultisols (1.3–1.5 Mg m⁻³), Alfisols (1.4–1.6 Mg m⁻³), and Gleysols (1.0–1.2 Mg m⁻³) (Batjes, 1996; Bernoux et al., 2002; Tomasella & Hodnett, 1998). Following conversion to agriculture, BD typically increases due to compaction and SOM depletion, reducing pore space and aggregate stability, which enhances microbial access to organic matter and accelerates SOC loss (Six et al., 2002; Lal, 2004). The magnitude of these changes varies by biome and soil type: sandy soils in the Cerrado and Caatinga show greater SOC sensitivity to BD increases, whereas clayey soils in the Atlantic Forest maintain partial resilience due to organo-mineral protection mechanisms. This extended discussion clarifies the biophysical mechanisms linking BD variability to SOC stock dynamics across tropical regions.

To support this point, we incorporated empirical evidence from Brazilian studies. Carvalho et al. (2010) and Maia et al. (2024) both report that conversion from native vegetation to agricultural or pastoral systems increases bulk density by approximately 0.13–0.20 Mg m⁻³. These changes reflect soil structural degradation caused by mechanical compaction and organic matter decline. This expanded discussion clarifies the biophysical mechanisms linking BD variability to SOC stock dynamics across tropical regions.

Reviewer #3 (Remarks to the Author):

The manuscript titled “Carbon debt in cropland soils of Brazil” presents a well-structured and timely contribution addressing one of the most pressing global environmental challenges—carbon sequestration and emissions associated with land use, particularly in the Brazilian context. Given Brazil’s pivotal role as both an agricultural powerhouse and a major player in global carbon dynamics, the findings are of high international relevance. The authors successfully identify carbon gaps across key Brazilian biomes, highlighting substantial opportunities for carbon restoration through land management strategies.

R: We sincerely thank the reviewer for this encouraging and insightful evaluation. We are pleased that the reviewer recognizes the structure, timeliness, and policy relevance of our study, as well as its contribution to understanding soil carbon dynamics and carbon gap distribution across Brazilian biomes.

Our primary objective was to provide a comprehensive and data-driven assessment of the carbon debt and recarbonization potential of Brazilian

croplands, framed within the global context of carbon neutrality and sustainable land management. We are encouraged that the reviewer found this approach relevant and aligned with international efforts to enhance soil carbon sequestration as part of climate change mitigation strategies.

We have carefully revised the manuscript based on all reviewer comments to further strengthen its analytical rigor, transparency, and applicability to both scientific and policy frameworks supporting carbon farming and MRV systems in tropical regions.

Major Comments:

Scientific Relevance and Novelty:

The study offers valuable insights by quantifying carbon debt across diverse biomes, reinforcing the potential for land management to mitigate carbon losses. The use of a comprehensive and current database strengthens the findings; however, the lack of data for certain biomes—especially the Pantanal—represents a notable limitation. A more robust discussion on this gap would enhance the study's completeness.

R: We appreciate the constructive feedback and fully agree that the scarcity of data for some biomes, especially the Pantanal, constitutes a significant limitation. In response, we expanded the Discussion section to acknowledge this gap and highlight the need for further empirical studies in underrepresented regions.

We observed that the low number of comparative SOC assessments in the Pantanal is associated with two factors: (i) limited agricultural expansion, as although soybean cultivation in grasslands has increased in recent years (Song et al., 2021), there are still few paired native and managed sites available; and (ii) logistical challenges, such as difficult access and seasonal flooding, which restrict long-term soil monitoring. These limitations increase uncertainty and reduce the representativeness of SOC gap estimates in this biome. We have included a new paragraph discussing these limitations and their implications for data coverage and uncertainty (Discussion, lines 164–170), highlighting those future efforts should prioritize data rescue, standardized sampling, and long-term monitoring in regions like the Pantanal to improve the spatial completeness of national SOC assessments.

Reference:

Song X-P., Hansen M. C., Potapov P. V., Adusei B., Pickering J., Adami M., Lima V., Zalles A., Stehman S. V., Di Bella C. M. et al. Massive soybean expansion in South

America since 2000 and implications for conservation. *Nature Sustainability* **4**, 784-792 (2021). DOI: 10.1038/s41893-021-00729-z.

Figures and Presentation:

The figures are generally well-designed and effectively support the narrative. However, there appears to be a discrepancy in figure numbering—specifically, there is no Figure 1. This should be corrected either by renumbering or including the missing figure. Additionally, several textual statements are not clearly linked to figures, and in some instances, the depth associated with reported results is missing. Clarifying these points will improve readability and data traceability.

R: We thank the reviewer for their careful review and positive comments regarding the figures. The numbering discrepancy has been corrected. Figure 1 (comparison between the general and regional approaches for the Cerrado biome) was included in the revised version, and all figures were renumbered and cross-checked throughout the manuscript to ensure consistency.

We also reviewed the entire text to ensure that each mention of visually presented results is clearly associated with the corresponding figure and that the related soil depths are consistently indicated in both the main text and the captions. These corrections improve traceability and consistency between figures and textual descriptions.

Language and Style:

Although the manuscript contains minor grammatical and stylistic errors, these do not detract significantly from the overall clarity. A careful revision by a native English speaker or a professional editing service is recommended to enhance fluency and polish.

R: We thank the reviewer for this helpful suggestion. In response, the entire manuscript has undergone a thorough language revision by a native English speaker with expertise in scientific writing. Minor grammatical and stylistic issues have been corrected to improve fluency, consistency, and overall readability. We believe that these revisions have enhanced the clarity and presentation quality of the manuscript.

Methodological Rigor:

The methodology appears sound; however, it is described too briefly in the current version. More detailed information in the Materials and Methods section is needed to allow for full reproducibility. For example, further clarification is required on how different land uses and management intensities were categorized and analyzed.

R: Thank you for your comment. We have updated the "Methods" section to include the new topic "Classification and Comparison of Management Systems," which provides a detailed description of how different soil types and management

intensities were categorized and analyzed. Additionally, the "Meta-analysis" topic has been revised to incorporate a description of the updated methodological procedure for stratified imputation (by biome, land use, and depth) of standard deviations in studies where these data were missing. These updates resulted in the inclusion of Table S28, which presents the median values of the coefficients of variation used to impute missing standard deviations, and Figure S8, which shows the results of the corresponding sensitivity analysis. Also in this section, we include a description of the methodological procedures for diagnosing heterogeneity, details of the models evaluated (random-effects model and mixed-effects model), a detailed approach by subgroup (latitude, temperature, rainfall, age of the agricultural system, and soil and climate classes), and a description of the procedure used to evaluate agricultural and management systems, based on a multilevel hierarchical model approach to address the issue of overlapping soil layers. Finally, we include the section "Sensitivity analyses and publication biases," which describes the funnel plot, Egger test, Fail-safe N (Rosenthal), and Trim and Fill statistical tests used to assess the sensitivity of the results to publication bias and the robustness of the analyses.

Title Accuracy:

The current title may be somewhat misleading, as the analysis extends beyond cropland to include grasslands and native soils. A more precise title would better reflect the manuscript's scope.

R: We appreciate the reviewer's thoughtful suggestion regarding title accuracy. We agree that the previous title could be interpreted as focusing exclusively on croplands, whereas our dataset and analysis encompass both croplands and grasslands in comparison to native vegetation.

To better reflect the true scope of the study, we have adopted the revised title:

“Soil carbon debt from land use change in Brazil.”

This new title captures the broader analytical framework while maintaining conciseness and scientific clarity, in line with the manuscript's objectives and *Nature Communications* style.

Conceptual Clarity:

The abstract and introduction would benefit from improved clarity. Several key topics—such as land use intensification, soil types, and climate analysis—are introduced abruptly or not at all in the early sections. For instance, three land uses are introduced initially, but five are later discussed, including grasslands,

which are not clearly integrated into the conceptual framework. These inconsistencies weaken the alignment between the study's objectives, methods, and results.

R: We thank the reviewer for this constructive comment. In response, we thoroughly revised both the *Abstract* and *Introduction* to improve clarity and ensure full conceptual alignment among the study's objectives, methods, and results. Specifically, we implemented the following changes:

1. We explicitly introduced all five land-use categories (monoculture, crop rotation, intercropping, grasslands, and integrated systems) at the beginning of the *Introduction*, thereby clearly integrating grasslands into the conceptual framework.
2. We clarified the roles of land-use intensification, soil types, and climatic variation in influencing SOC dynamics, explicitly linking these factors to the study's objectives and methodological structure.
3. We refined the *Abstract* to reflect the full range of land uses and analytical scope, providing a more accurate and coherent summary of the study's goals, methods, and key findings.
4. These revisions strengthen the logical coherence and readability of the manuscript, ensuring that the conceptual framework is consistently represented throughout the *Abstract*, *Introduction*, and subsequent sections.

Management Trade-Offs:

One critical issue that warrants deeper discussion is the dual role of agricultural intensification. While such practices can enhance soil carbon stocks under certain conditions, they can also lead to negative soil quality outcomes—such as compaction—particularly due to increased machinery traffic in high-frequency crop rotations. Additionally, it should be emphasized that intensification strategies may only be sustainable on soils with high agricultural suitability; applying them to fragile or marginal lands could exacerbate degradation.

R: We thank the reviewer for raising this important point. We agree that agricultural intensification entails both opportunities and risks for SOC and overall soil health. We have now expanded the Discussion (lines 415–421) to provide a more balanced view of this dual role. The revised text clarifies that, under favorable conditions and proper management, such as increased residue return, crop diversification, and reduced soil disturbance, intensification can enhance SOC accumulation and soil structure. However, when implemented

unsustainably, especially in fragile or marginal soils, or with excessive machinery traffic and high cropping frequency, intensification may increase bulk density, reduce porosity, and accelerate carbon losses.

We also highlight that intensification strategies should be prioritized in soils with high agricultural suitability, whereas applying them to degraded or low-resilience soils can exacerbate compaction and long-term soil degradation.

Minor Comments:

Ensure that all statements in the Results section are directly supported by corresponding figures and data.

R: We carefully reviewed the Results section to confirm that all statements are now directly supported by figures, tables, or supplementary data. Clarify the depth of soil sampling where it is not currently specified.

R: We verified and standardized the indication of soil depths throughout the manuscript and in all figure captions to ensure consistency and traceability. Detailed comments are noted in the Manuscript pdf.

R: All detailed comments were addressed individually in the revised version. Minor grammatical, formatting, and cross-referencing issues were corrected throughout the manuscript.

Line 27 - I'm not use to this format. But here you don't have to include a subtitle?

R: The subtitle was removed following the journal's formatting guidelines.

Line 35 - You must describe the meaning of low-C agriculture.

R: This content has been removed from the revised manuscript; therefore, this comment no longer applies.

Line 52 - Putt (-) next to 7 in the next line;

R: Correction implemented as suggested.

Line 52 - Decide according the journal style if you are going to leave a space between Fig and the number. When you address supplementary Table you have left a gap;

R: All references to figures and supplementary tables have been standardized according to the *Nature Communications* style guide.

Line 62 - and others variables, biophysical, climate, management, etc.

R: This clarification has been added to the text.

Line 77 - Fig, 2a

R: Corrected as suggested.

Line 78 - cm;

R: Unit of measurement corrected and standardized.

Line 93 – For the whole country? Because you are analysing Brazilian biomes;

R: Clarified in the text that the analysis refers to Brazilian biomes rather than the entire national territory.

Line 95 – Rephrase

R: Sentence rephrased for clarity and conciseness.

Figure 2 comments - (Where is figure 1?)

R: The figure numbering discrepancy was corrected; Figure 1 has been added and all figures renumbered accordingly.

Line 102 – From here to the next paragraphs its not clear the depth that you are describing

R: Soil depth specification was added throughout this section to improve clarity.

Line 105 – For all biomes?

R: Clarified in the text that the result refers to all six Brazilian biomes.

Line 106 – this is crop rotation (crops and pasture/cattle) or an intensified cropping system, e.g two or more crops?

R: Clarified in the methods section that the category refers to intensified cropping systems involving two or more crop cycles.

Line 107 – You named differently at the beginning of the study. Is intercropping or a more intensified crop sequence.

R: Terminology standardized across the manuscript for consistency.

Line 114 – You did not present the Grassland classification before;

R: Grassland classification is now introduced earlier in the *Introduction* and detailed in the *Methods* section.

Line 115 – There's a gap here;

R: Sentence completed to ensure narrative continuity.

Line 117 – Table?

R: Reference to the relevant table added.

Line 118 – same comment as Grassland

R: The term was harmonized across sections to maintain consistent nomenclature.

Line 122 – Here you mention 5 land uses not introduced together at the beginning of the study

R: The *Introduction* was revised to introduce all five land-use categories upfront.

Line 127 – see if the draw backs of intensification is latter addressed

R: The potential trade-offs of intensification, including risks of compaction and degradation, were discussed in the *Discussion* section.

Line 135 – Here it is crucial to know the depths involved due to stratification of NT

R: We thank the reviewer for this valuable technical comment. To address this point, we clarified that the assessment of agricultural and management systems (no-tillage and conventional tillage) jointly encompassed the 0–10 cm, 0–20 cm, and 0–30 cm soil layers. We also conducted an additional analysis evaluating the effect by individual layer. The results are now presented as Figure S5 in the Supplementary Material, and a corresponding sentence has been added to the main text to describe the outcomes.

Figure S5. Assessment of the stratification effect in the no-tillage system across soil layers.

The following sentence has been added to the main text.

“We also compared SOC_{stocks} losses from NV to no-tillage and conventional tillage systems. Our results confirmed that losses under no-tillage (-11.4%, CI [-15.7, -6.9%]) were 47% lower than under conventional tillage (-21.4%, CI [-25.4, -17.3]). The Pampa and Atlantic Forest recorded the most significant losses under the conventional system, respectively -32.7% and -27.2%. The loss in the Pampa was 88–110% greater than in the Cerrado (-17.2%), Caatinga (-17.4%), and Amazon (-15.6%) biomes, while in the Atlantic Forest it exceeded these biomes by 58–74%. Under the no-tillage system, the Atlantic Forest recorded the greatest loss of SOC_{stocks} (-21.9%), followed by the Pampa (-10.6%), while the Cerrado and Amazon presented the smallest losses of -8.0% and -7.4%, respectively. We also evaluated the effect of the no-tillage system on different soil depths to identify possible variations associated with the stratification induced by this management. The results (Supplementary Fig. S5) indicated a small variation between the original mean value (-11.4%) and those obtained in the stratified analysis (0–10 cm = -14.0%; 0–20 cm and 0–30 cm ≈ -

10.0%), with a more pronounced difference in the 0–10 cm layer and slightly smaller ones in the subsequent layers.”

Line 161 – Perhaps, I missed but I believe that this section is related to Fig 3, but it’s not cited in the text.

R: We included the reference to Figure 3 in the text, ensuring the traceability of the information presented in the respective sentence.

It difficult to read. Needs more contrast/Here you include the name again ad not before;

R: We thank the reviewer for this observation. In response, we have adjusted the figure to improve contrast and readability. Additionally, we have corrected the placement of the label, ensuring that the name appears consistently and clearly at the appropriate location.

Line 224 - studies?

R: We thank the reviewer for noting this issue. The sentence at line 224 has been revised for grammatical accuracy and clarity. It now correctly refers to “studies” in the plural form, ensuring agreement and precision in the description of the referenced works.

Line 234 – you may put these in extent because not everybody remember those acronyms

R: We thank the reviewer for the comment. We have added the description of the climate classes corresponding to the acronyms mentioned in the text. The updated sentence inserted in the manuscript is presented below.

“More specifically, SOC_{stocks} under NV and AGR in most soil layers of the biomes occurred in the following climatic order: Tropical rainforest-Af, Temperate (no dry season/warm summer)-Cfb and Temperate (dry winter/warm summer)-Cwb > Temperate (dry winter/hot summer)-Cwa, Temperate (no dry season/hot summer)-Cfa, Tropical Monsoon-Am, Tropical Savannah (summer rain)-Aw, Tropical Savannah (winter rain)-As > Arid (steppe/ hot)-BSh (Supplementary Tables S10,11).”

Line 237 – The following paragraphs regarding soil types and carbon stock were not mentioned as objectives nor mentioned in the M&M

R: We thank the reviewer for the comment. We have included information on soil and climate classes in the statement of the study objective (lines 48–60).

Line 237 – Include what you called topsoil. It differs from each soil type?/ of the evaluated biomes?

R: We thank the reviewer for the comment. We have revised the sentence to clarify that the analysis considered the available SOC_{stocks} data across all biomes for the five main soil classes, as well as the definition of *topsoil* adopted in this study, which includes soil layers of 0–10 cm, 0–20 cm, and 0–30 cm depth (lines 119–128).

Figure 4 – comments

Unit

R: We thank the reviewer for the comment. The figure was updated to include the SOC_{stock} unit (Mg ha⁻¹) on both the upper and lower x-axes. The reorganization of the Results sections required the renumbering of “Figure 4”, which is now presented as “Figure 1”.

At first glance I did not understand this Figure, because I did not realize that the depth was accumulative, perhaps you can include same statement regarding this.

R: We thank the reviewer for this helpful observation. We agree that the figure could be misinterpreted as representing cumulative soil depths. To clarify this, we explicitly stated in the figure caption that the values for each depth layer are independent and should not be interpreted cumulatively.

This clarification ensures a correct understanding that the results represent separate depth intervals, not accumulated SOC stocks across the soil profile (lines 493–495).

Line 317 – Include them here

R: We thank the reviewer for the comment. The sentence has been updated with the aforementioned suggestion.

Line 334 – put “s” on soc stock

R: We replaced “SOC_{stocks}” with “SOC_{stocks}”

Line 364 – Before you only use the uppercase number;

R: The reference citation has been adjusted to comply with the Nature Communications formatting guidelines.

Line 378 – Before you use P

R: We standardized the reporting of p-values as $p < 0.05$ or 0.001 .

Line 379 – Here would be interesting comment what were you findings...

R: We revised the Methods section by moving publication bias test details from 'Meta-analysis' to a new section, 'Sensitivity analyses and publication biases.' This new section covers the Funnel Plot and Egger tests as well as the newly added Fail-safe N (Rosenthal) analyses. We also included a summarizing sentence at the end of the section about the results of the statistical analyses assessing robustness. Below, we present the updated content of the 'Sensitivity analyses and publication biases' section.

“We assessed the potential for publication bias (Supplementary Fig. S5 and S6) using the funnel plot and Egger’s test⁹³. The test assesses the existence of publication bias through regression analysis using the effect size and its precision (standard error). A significant intercept (indexed by b), i.e., with $p < 0.05$, suggests publication bias⁹⁴. To assess the sensitivity of the results to publication bias, Rosenthal’s fail-safe number⁹⁵ was applied, which estimates the number of hypothetical studies with a null effect required to nullify the significance of the combined effect (Supplementary Table S28 and Fig. S8). The trim-and-fill procedure was also used to adjust the mean effect value and thus take possible missing studies into account. The findings from the combination of the three methods confirm that the results of the meta-analyses by biome are, for the most part, statistically robust and little affected by publication bias. The extremely high Rosenthal’s fail-safe N values and the absence of trim-and-fill adjustments reinforce the reliability of the estimates. The Cerrado showed the greatest sensitivity to potential bias in the 0–10 cm layer, but, even in this case, the overall effect remained significant and stable. It is worth noting that methods such as Egger’s test and the funnel plot have their own limitations, especially in contexts with high heterogeneity of the observed effects⁹⁶, which applies to our study.”

Line 394 – Wich variables?

R: Clarified in the text which variables are being referred to. The sentence now specifies the variables included in the model.

Line 660 – Gaps

R: The term “gaps” was clarified and standardized throughout the text to ensure consistent terminology

Line 665 – year

R: The time reference was revised for accuracy and consistency.

Line 668 - complete

R: The sentence was completed and rephrased to improve clarity.

Referencias

Kambach, S., Bruelheide, H., Gerstner, K., Gurevitch, J., Beckmann, M. & Seppelt, R. Consequences of multiple imputation of missing standard deviations and sample sizes in meta-analysis. *Ecol. Evol.* **10**, 11699–11712 (2020).

Davison, A. C. & Hinkley, D. V. *Bootstrap Methods and their Application*. (Cambridge Univ. Press, 1997).

Table1. Results of the meta-analysis by biome and soil layer, including effect size estimates, confidence intervals, and heterogeneity statistics, as well as the explanatory power of the moderators (*Conv_Time* and *Time_span*).

Biome	Soil layer (cm)	Term	Estimate	SE	Zval	Pval	CI.lb	CI.ub	Model	τ^2	I ²	H ²	QE	QEp	R ² (%)
Pampa	0 - 10	intcpt	-5.4	0.8	-6.4	1.41E-10	-7.11	-3.78	Null	16.0	90.8	10.9	255.2	5E-40	
		Conv_Time	-0.8	1.0	-0.8	4.00E-01	-2.66	1.07	Moderators	16.7	91.3	11.4	232.5	1.4E-36	-4.7
		Time_span	-0.3	0.9	-0.3	7.74E-01	-2.09	1.55							
	0 - 20	intcpt	-7.4	1.0	-7.3	3.40E-13	-9.35	-5.38	Null	82.2	100.0	7684.2	5981.5	0	
		Conv_Time	2.1	1.1	1.9	5.57E-02	-0.05	4.24	Moderators	72.9	100.0	6992.4	5034.5	0	11.4
		Time_span	-2.2	1.1	-2.0	4.21E-02	-4.30	-0.08							
	0 - 30	intcpt	-6.2	1.5	-4.1	3.48E-05	-9.20	-3.29	Null	78.9	98.1	52.6	724.7	2.8E-138	
		Conv_Time	-3.7	1.5	-2.4	1.48E-02	-6.74	-0.73	Moderators	49.7	97.1	34.6	452.2	1.35E-82	37.0
		Time_span	4.0	1.5	2.6	8.77E-03	1.01	7.03							
Pantanal	0 - 10	intcpt	-3.9	0.9	-4.4	1.09E-05	-5.66	-2.17	Null	4.6	89.1	9.2	42.1	5.7E-08	
		Conv_Time	-47.4	31.5	-1.5	1.33E-01	-109.26	14.38	Moderators	4.0	88.7	8.9	17.9	0.0005	12.9
		Time_span	47.1	31.6	1.5	1.36E-01	-14.79	108.94							
	0 - 20	intcpt	-4.8	2.1	-2.3	2.31E-02	-8.98	-0.66	Null	18.1	97.5	40.0	162.0	3.68E-33	
		Conv_Time	-39.1	73.1	-0.5	5.92E-01	-182.30	104.06	Moderators	26.4	98.3	60.3	94.8	2.01E-20	-45.7
		Time_span	38.1	73.1	0.5	6.02E-01	-105.11	181.34							
	0 - 30	intcpt	-5.3	0.5	-10.8	4.85E-27	-6.27	-4.34	Null	10.6	89.9	9.9	37.4	1.47E-07	
		Conv_Time	49.4	9.2	5.3	8.80E-08	31.27	67.43	Moderators	0.0	1.4	1.0	2.0	0.37	99.8
		Time_span	-50.5	9.2	-5.5	3.95E-08	-68.55	-32.50							
Amazon	0 - 10	intcpt	-4.0	0.6	-6.2	7.47E-10	-5.21	-2.69	Null	4.2	52.9	2.1	50.5	0.0012	
		Conv_Time	-2.3	0.9	-2.5	1.11E-02	-4.00	-0.51	Moderators	2.8	44.2	1.8	38.8	0.014	32.8
		Time_span	2.3	0.9	2.5	1.42E-02	0.47	4.17							
	0 - 20	intcpt	-4.2	1.0	-4.2	3.09E-05	-6.18	-2.23	Null	29.0	91.8	12.1	313.6	1.83E-47	
		Conv_Time	4.4	3.5	1.2	2.16E-01	-2.54	11.25	Moderators	29.0	91.7	12.1	297.3	2.8E-45	-0.2
		Time_span	-3.3	3.5	-1.0	3.42E-01	-10.19	3.54							
	0 - 30	intcpt	-2.9	0.9	-3.2	1.47E-03	-4.73	-1.12	Null	61.4	99.2	120.2	3528.4	0	
		Conv_Time	-1.6	1.0	-1.6	1.18E-01	-3.55	0.40	Moderators	51.6	98.9	92.0	2293.8	0	15.9
		Time_span	3.6	1.0	3.5	4.33E-04	1.57	5.53							

-Continued on next page-

Caatinga	0 - 10	intrcpt	-2.2	1.0	-2.2	3.10E-02	-4.22	-0.20	Null	45.5	95.8	23.8	792.7	1.3E-138	
		Conv_Time	-0.1	1.2	-0.1	9.00E-01	-2.41	2.11	Moderators	41.9	95.3	21.2	659.5	1.5E-112	7.9
		Time_span	-2.2	1.1	-2.0	4.73E-02	-4.47	-0.03							
	0 - 20	intrcpt	-4.8	2.1	-2.3	2.31E-02	-8.98	-0.66	intrcpt	-4.8	2.1	-2.3	0.0	-8.9	-0.7
		Conv_Time	-39.1	73.1	-0.5	5.92E-01	-182.30	104.06	Conv_Time	-39.1	73.1	-0.5	0.6	-182.3	
		Time_span	38.1	73.1	0.5	6.02E-01	-105.11	181.34	Time_span	38.1	73.1	0.5	0.6	-105.11	
	0 - 30	intrcpt	-4.5	2.1	-2.2	3.02E-02	-8.65	-0.44	Null	121.0	97.9	47.1	828.1	3.9E-160	
		Conv_Time	2.7	2.1	1.2	2.11E-01	-1.51	6.81	Moderators	100.2	97.2	35.7	612.5	3.8E-116	17.2
		Time_span	4.6	2.1	2.1	3.19E-02	0.40	8.73							
Cerrado	0 - 10	intrcpt	-4.0	0.4	-9.3	2.00E-20	-4.87	-3.17	Null	49.5	97.7	44.1	10028.1	0	
		Conv_Time	-0.6	0.4	-1.5	1.41E-01	-1.50	0.21	Moderators	49.4	97.7	42.7	9655.7	0	0.3
		Time_span	-0.2	0.4	-0.5	5.83E-01	-1.11	0.62							
	0 - 20	intrcpt	-7.3	0.7	-10.8	4.01E-27	-8.57	-5.93	Null	94.7	98.5	66.7	14772.6	0	
		Conv_Time	0.1	0.7	0.1	9.14E-01	-1.27	1.42	Moderators	95.1	98.4	61.9	14583.7	0	-0.5
		Time_span	0.7	0.7	1.0	3.08E-01	-0.65	2.05							
	0 - 30	intrcpt	-5.8	0.6	-9.1	6.50E-20	-7.03	-4.55	Null	51.7	97.9	46.9	3501.2	0	
		Conv_Time	0.5	0.7	0.8	4.52E-01	-0.82	1.84	Moderators	52.2	97.7	42.9	3407.7	0	-1.0
		Time_span	0.5	0.6	0.8	3.99E-01	-0.71	1.79							
Atlantic Forest	0 - 10	intrcpt	-5.5	0.8	-6.8	8.35E-12	-7.10	-3.93	Null	61.2	99.1	116.7	4807.2	0	
		Conv_Time	-0.1	0.9	-0.1	9.46E-01	-1.76	1.64	Moderators	62.3	99.1	117.1	4665.4	0	-1.7
		Time_span	-0.5	0.9	-0.6	5.47E-01	-2.23	1.18							
	0 - 20	intrcpt	-7.6	1.0	-7.5	5.12E-14	-9.53	-5.60	Null	90.1	97.1	34.1	2827.6	0	
		Conv_Time	-2.6	1.0	-2.5	1.21E-02	-4.61	-0.57	Moderators	85.3	96.9	31.9	2478.6	0	5.4
		Time_span	1.1	1.0	1.1	2.88E-01	-0.94	3.17							
	0 - 30	intrcpt	-7.1	1.4	-5.0	4.79E-07	-9.85	-4.33	Null	130.2	98.3	59.2	2613.4	0	
		Conv_Time	-1.5	1.6	-0.9	3.48E-01	-4.59	1.62	Moderators	131.3	98.3	58.6	2485.1	0	-0.9
		Time_span	-0.5	1.6	-0.3	7.32E-01	-3.61	2.54							

Biome = biome considered; **Soil layer (cm)** = depth of the soil layer evaluated; **Term** = parameter estimated in the model; **Estimate** = point estimate of the effect; **SE** = standard error of the estimate; **Zval** = Z statistic value; **Pval** = p-value associated with the test; **CI.lb / CI.ub** = lower and upper bounds of the 95% confidence interval; **Model** = fitted model type (random- or mixed-effects); τ^2 (**tau²**) = between-study variance (unexplained heterogeneity); **I²** = proportion of total variability due to heterogeneity; **H²** = ratio of total to sampling variance (relative heterogeneity); **QE** = Q statistic for residual heterogeneity; **QE_p** = p-value associated with the QE test; **R² (%)** = proportion of heterogeneity explained by moderators.

Appendix I Meta-analysis results by biome and soil layer, showing effect estimates, confidence intervals, and heterogeneity and moderator-explained variance metrics.

Biome	Soil layer (cm)	Term	Estimate	SE	Zval	Pval	CI.lb	CI.ub	Model	τ^2	I ²	H ²	QE	QEp	R ² (%)
Cerrado	0 - 10	intrcpt	1.8	5.1	0.4	0.720	-8.1	11.7	Null	49.5	97.7	44.1	10,028	0	17.0
		Temperature	0.3	0.6	0.4	0.685	-1.0	1.5	Moderators	41.1	97.0	33.8	6,576	0	
		Rainfall	0.4	0.7	0.5	0.628	-1.1	1.8							
		Lat	2.7	0.8	3.6	0.000	1.2	4.2							
		long	0.3	0.9	0.4	0.695	-1.3	2.0							
		Conv_Time_z	-0.1	0.4	-0.2	0.850	-1.0	0.8							
		Time_span_z	0.1	0.4	0.2	0.817	-0.8	1.0							
		SoilCInceptisols	-17.1	5.0	-3.4	0.001	-26.8	-7.4							
		SoilCOxisols	-6.2	2.5	-2.5	0.014	-11.1	-1.3							
		SoilCULTisols	-7.7	3.0	-2.6	0.010	-13.5	-1.8							
		CSEucalyptus	-2.6	2.1	-1.2	0.212	-6.7	1.5							
		CSGrassland	1.5	1.3	1.1	0.261	-1.1	4.2							
		CSIAS	0.6	1.3	0.5	0.643	-1.9	3.1							
		CSMinimum-Tillage	-1.8	3.6	-0.5	0.624	-8.8	5.3							
		CSNo-Tillage	1.2	1.1	1.1	0.271	-1.0	3.4							
		CSPinus	2.1	6.7	0.3	0.750	-11.0	15.3							
		ClimateAm	6.7	4.8	1.4	0.161	-2.7	16.0							
		ClimateAs	0.3	4.8	0.1	0.946	-9.2	9.8							
		ClimateAw	-0.7	4.1	-0.2	0.865	-8.7	7.3							
		ClimateCwa	2.8	5.0	0.6	0.575	-7.0	12.6							
ClimateCwb	2.0	6.3	0.3	0.755	-10.3	14.2									
Cerrado	0 - 20	intrcpt	-27.9	14.1	-2.0	0.047	-55.5	-0.3	Null	94.7	98.5	66.7	14773	0	
		Temperature_z	-0.6	1.0	-0.6	0.562	-2.7	1.4	Moderators	81.6	98.0	50.2	8536	0	
		Rainfall_z	-2.6	1.1	-2.4	0.018	-4.8	-0.5							
		Lat_z	3.5	1.3	2.8	0.006	1.0	5.9							
		long_z	-1.5	1.3	-1.1	0.252	-4.0	1.1							
		Conv_Time_z	0.2	0.7	0.3	0.765	-1.2	1.6							
		Time_span_z	1.1	0.7	1.6	0.109	-0.2	2.4							
		SoilCEntisols	38.6	13.8	2.8	0.005	11.5	65.7							
		SoilCInceptisols	22.1	14.0	1.6	0.114	-5.3	49.4							
		SoilCOxisols	30.0	13.2	2.3	0.024	4.0	55.9							

SoilCultisols	26.9	13.5	2.0	0.046	0.4	53.4
CSEucalyptus	-4.4	3.2	-1.4	0.164	-10.6	1.8
CSGrassland	-3.7	2.1	-1.8	0.078	-7.8	0.4
CSIAS	0.2	2.3	0.1	0.923	-4.3	4.8
CSMinimum-Tillage	9.6	5.1	1.9	0.063	-0.5	19.6
CSNo-Tillage	1.2	1.9	0.6	0.524	-2.5	4.9
CSPerennial	-8.1	9.7	-0.8	0.404	-27.0	10.9
CSPinus	2.1	9.6	0.2	0.825	-16.7	21.0
CSReduced Tillage	11.5	9.4	1.2	0.220	-6.9	29.8
ClimateAw	-9.0	4.5	-2.0	0.047	-17.9	-0.1
ClimateCwa	-6.4	5.2	-1.2	0.224	-16.6	3.9
ClimateCwb	-4.4	11.0	-0.4	0.693	-26.0	17.3

	intrcpt	3.2	7.2	0.4	0.662	-11.0	17.3	Null	51.7	97.9	46.93	3501	0	
	Temperature_z	0.4	1.1	0.4	0.688	-1.7	2.6	Moderators	44.6	96.7	30.59	2023	0	13.7
	Rainfall_z	2.8	1.1	2.5	0.014	0.6	5.0							
	Lat_z	1.2	1.3	0.9	0.359	-1.4	3.8							
	long_z	3.1	1.3	2.4	0.015	0.6	5.6							
	Conv_Time_z	1.8	0.9	2.0	0.045	0.0	3.6							
	Time_span_z	0.6	0.6	0.9	0.353	-0.7	1.8							
	SoilCInceptisols	-13.8	9.0	-1.5	0.123	-31.4	3.8							
	SoilCOxisols	-1.8	3.1	-0.6	0.561	-7.8	4.2							
0 - 30	SoilCultisols	-14.4	6.6	-2.2	0.029	-27.3	-1.5							
	CSEucalyptus	-5.9	3.2	-1.8	0.065	-12.1	0.4							
	CSFallow	2.0	7.4	0.3	0.790	-12.5	16.4							
	CSGrassland	4.4	1.9	2.3	0.023	0.6	8.2							
	CSIAS	5.3	3.5	1.5	0.132	-1.6	12.2							
	CSNo-Tillage	0.6	1.6	0.4	0.695	-2.5	3.8							
	CSPinus	0.0	7.5	0.0	0.995	-14.7	14.8							
	ClimateAm	-1.1	7.3	-0.2	0.879	-15.3	13.1							
	ClimateAw	-8.5	6.6	-1.3	0.198	-21.5	4.5							
	ClimateCfa	7.1	10.1	0.7	0.486	-12.8	26.9							
	ClimateCwa	-8.8	7.5	-1.2	0.236	-23.5	5.8							
	ClimateCwb	-8.9	11.0	-0.8	0.421	-30.4	12.7							

Atlantic Forest

0 - 10	intrcpt	-8.6	6.3	-1.4	0.170	-20.9	3.7	Null	61.2	99.1	116.7	4807	0	
	Temperature_z	5.3	2.2	2.4	0.019	0.9	9.7	Moderators	39.8	98.6	73.5	3022	0	35.0
	Rainfall_z	2.1	1.3	1.6	0.108	-0.5	4.6							

	Lat_z	1.9	4.3	0.4	0.668	-6.6	10.3							
	Conv_Time_z	1.8	0.9	2.0	0.048	0.0	3.6							
	Time_span_z	-1.1	0.9	-1.1	0.253	-2.9	0.8							
	Long_z	-1.6	2.2	-0.7	0.471	-6.0	2.8							
	SoilCOxisols	-0.5	3.0	-0.2	0.866	-6.3	5.3							
	SoilCULTisols	-3.3	3.9	-0.9	0.390	-11.0	4.3							
	CSGrassland	6.1	2.5	2.5	0.014	1.2	10.9							
	CSIAS	3.6	4.2	0.8	0.396	-4.7	11.9							
	CSNo-Tillage	2.8	1.9	1.5	0.145	-1.0	6.5							
	CSPerennial	-0.5	3.9	-0.1	0.898	-8.1	7.1							
	ClimateAs	-1.2	5.8	-0.2	0.843	-12.5	10.2							
	ClimateAw	2.2	5.5	0.4	0.695	-8.7	13.0							
	ClimateCfa	-1.6	6.8	-0.2	0.810	-14.9	11.6							
	ClimateCfb	7.3	7.2	1.0	0.308	-6.7	21.4							
	ClimateCwa	3.6	5.9	0.6	0.546	-8.1	15.2							
	ClimateCwb	9.6	7.5	1.3	0.205	-5.2	24.4							
	intrcpt	8.1	15.1	0.5	0.595	-21.6	37.7	Null	90.1	97.1	34.1	2828	0	
	Temperature_z	2.9	2.8	1.0	0.295	-2.6	8.4	Moderators	58.7	95.1	20.3	1590	8E-286	34.90
	Rainfall_z	5.5	3.2	1.7	0.082	-0.7	11.7							
	Lat_z	-0.5	3.9	-0.1	0.904	-8.1	7.1							
	Conv_Time_z	1.3	1.5	0.8	0.399	-1.7	4.2							
	Time_span_z	-0.2	1.5	-0.2	0.880	-3.1	2.7							
	Long_z	5.3	3.2	1.7	0.098	-1.0	11.6							
	SoilCOxisols	9.7	3.3	2.9	0.003	3.3	16.2							
	SoilCULTisols	-10.4	8.4	-1.2	0.214	-26.9	6.0							
0 - 20	CSGrassland	8.1	3.6	2.2	0.025	1.0	15.1							
	CSIAS	-11.6	8.2	-1.4	0.157	-27.8	4.5							
	CSNo-Tillage	5.6	2.1	2.6	0.008	1.4	9.7							
	CSPerennial	0.9	16.5	0.1	0.957	-31.5	33.3							
	CSSilviculture	5.1	9.4	0.5	0.590	-13.4	23.6							
	ClimateAs	-15.3	9.0	-1.7	0.090	-33.0	2.4							
	ClimateAw	-36.6	16.8	-2.2	0.029	-69.5	-3.7							
	ClimateCfa	-26.2	15.4	-1.7	0.089	-56.5	4.0							
	ClimateCfb	-26.6	15.5	-1.7	0.085	-57.0	3.7							
	ClimateCwa	-21.1	14.6	-1.4	0.147	-49.7	7.5							
	ClimateCwb	-25.1	15.2	-1.6	0.099	-54.9	4.7							
0 - 30	intrcpt	-24.8	18.9	-1.3	0.189	-61.7	12.2	Null	130.2	98.3	59.2	2613	0	

	Temperature_z	9.1	4.8	1.9	0.058	-0.3	18.5	Moderators	96.1	97.4	39.0	1557	3E-293	26.2
	Rainfall_z	9.1	4.2	2.2	0.031	0.8	17.4							
	Lat_z	-2.3	7.9	-0.3	0.775	-17.8	13.3							
	Conv_Time_z	2.0	2.0	1.0	0.321	-1.9	5.8							
	Time_span_z	-2.4	1.8	-1.3	0.180	-5.8	1.1							
	Long_z	9.2	8.8	1.0	0.297	-8.1	26.5							
	SoilCOxisols	13.1	5.0	2.6	0.009	3.3	22.9							
	SoilCULTisols	-1.5	9.9	-0.1	0.883	-20.8	17.9							
	CSGrassland	12.5	6.0	2.1	0.035	0.9	24.2							
	CSIAS	13.6	7.3	1.9	0.061	-0.6	27.9							
	CSNo-Tillage	9.6	4.3	2.2	0.026	1.2	18.1							
	CSPerennial	-25.3	21.1	-1.2	0.232	-66.7	16.1							
	ClimateAs	-12.8	12.2	-1.0	0.294	-36.7	11.1							
	ClimateAw	-5.8	19.7	-0.3	0.769	-44.4	32.8							
	ClimateCfa	3.4	21.6	0.2	0.874	-38.9	45.8							
	ClimateCfb	1.4	19.7	0.1	0.943	-37.3	40.1							
	ClimateCwa	4.8	15.1	0.3	0.749	-24.7	34.4							
	ClimateCwb	15.7	17.8	0.9	0.378	-19.3	50.7							
	intrcpt	54.3	68.0	0.8	0.42461	-79.0	187.6	Null	4.2	52.9	2.1	50.5	0.001	
	Temperature_z	-0.9	3.4	-0.3	0.78207	-7.6	5.7	Moderators	1.2	23.6	1.3	15.6	0.272	71.2
	Rainfall_z	-72.2	78.8	-0.9	0.35968	-226.6	82.2							
	Lat_z	10.3	20.3	0.5	0.61156	-29.5	50.2							
	Conv_Time_z	-5.6	12.6	-0.4	0.65549	-30.4	19.1							
0 - 10	Time_Span_z	2.7	1.6	1.7	0.09515	-0.5	6.0							
	Long_z	-65.8	72.7	-0.9	0.36513	-208.3	76.6							
	CSFallow	8.5	5.5	1.5	0.12374	-2.3	19.4							
	CSGrassland	2.9	4.2	0.7	0.49107	-5.3	11.1							
	CSIAS	3.6	2.7	1.3	0.19066	-1.8	9.0							
	CSNo-Tillage	2.4	1.4	1.8	0.07816	-0.3	5.0							
	SoilCOxisol	-66.2	74.9	-0.9	0.37673	-213.1	80.6							
	intrcpt	-9.2	4.6	-2.0	0.04769	-18.2	-0.1	Null	22.9	90.2	10.2	281.9	3E-42	
	Temperature_z	12.5	6.2	2.0	0.04415	0.3	24.6	Moderators	16.9	86.8	7.6	164.0	1E-24	26.0
	Rainfall_z	-12.3	6.6	-1.9	0.06112	-25.3	0.6							
0 - 20	Lat_z	22.0	9.8	2.3	0.02390	2.9	41.1							
	Conv_Time_z	5.2	6.6	0.8	0.42589	-7.7	18.1							
	Time_Span_z	-3.6	6.0	-0.6	0.55576	-15.4	8.3							
	Long_z	-27.0	11.3	-2.4	0.01727	-49.1	-4.8							

		CSGrassland	7.4	5.2	1.4	0.15536	-2.8	17.6							
		CSIAS	6.3	4.4	1.5	0.14643	-2.2	14.9							
		CSNo-Tillage	4.9	4.1	1.2	0.22734	-3.1	12.9							
		SoilCULTisol	0.0	4.9	0.0	0.99889	-9.7	9.7							
		ClimateAw	-5.7	4.2	-1.4	0.16983	-13.9	2.5							
		intrcpt	11.5	6.7	1.7	0.08771	-1.7	24.7	Null	61.4	99.2	120.2	3528.4	0	
		Temperature_z	-5.5	1.5	-3.6	0.00028	-8.4	-2.5	Moderators	25.0	97.5	40.4	1317.3	5E-240	59.3
		Rainfall_z	0.9	1.1	0.8	0.44494	-1.4	3.1							
		Lat_z	6.0	1.6	3.7	0.00026	2.8	9.2							
		Conv_Time_z	-3.0	1.4	-2.1	0.03382	-5.8	-0.2							
		Time_Span_z	1.6	1.1	1.4	0.15501	-0.6	3.8							
		Long_z	-5.7	1.3	-4.4	0.00001	-8.2	-3.1							
	0 - 30	CSGrassland	7.7	4.1	1.9	0.05892	-0.3	15.7							
		CSIAS	8.1	4.0	2.0	0.04295	0.3	16.0							
		CSNo-Tillage	4.8	4.2	1.2	0.24964	-3.4	12.9							
		CSPerennial	-4.9	4.5	-1.1	0.27041	-13.7	3.8							
		CSSilviculture	4.3	5.6	0.8	0.44602	-6.7	15.3							
		SoilCOxisol	-9.8	5.0	-1.9	0.05264	-19.6	0.1							
		SoilCULTisol	2.4	7.1	0.3	0.73060	-11.4	16.3							
		ClimateAm	-10.4	4.6	-2.3	0.02323	-19.3	-1.4							
		ClimateAw	-14.0	5.2	-2.7	0.00725	-24.3	-3.8							
		intrcpt	-12.4	8.4	-1.5	0.138	-28.8	4.0	Null	45.5	95.8	23.8	792.7	1E-138	
		Temperature_z	1.1	2.0	0.5	0.597	-2.9	5.1	Moderators	41.2	94.5	18.2	399.8	3E-69	9.6
		Rainfall_z	-1.0	1.8	-0.5	0.584	-4.4	2.5							
		Lat_z	-2.8	3.6	-0.8	0.434	-9.8	4.2							
		Time_span_z	-2.7	2.7	-1.0	0.313	-7.9	2.5							
		Conv_Time_z	-3.4	3.5	-1.0	0.334	-10.3	3.5							
		Long_z	2.7	2.7	1.0	0.326	-2.7	8.1							
	0 - 10	CSGrassland	4.7	3.7	1.3	0.204	-2.6	12.1							
		CSIAS	4.9	5.8	0.8	0.397	-6.4	16.3							
Caatinga		CSminimum tillage	4.1	9.4	0.4	0.660	-14.2	22.5							
		CSPerennial	5.3	15.0	0.4	0.724	-24.1	34.7							
		CSSilviculture	0.6	8.3	0.1	0.939	-15.5	16.8							
		SoilCENTisol	6.6	6.5	1.0	0.309	-6.2	19.5							
		SoilCInceptisol	2.4	7.5	0.3	0.746	-12.3	17.2							
		SoilCInceptisols	5.9	6.1	1.0	0.333	-6.1	17.9							
		SoilCoxisol	5.7	11.5	0.5	0.623	-16.9	28.2							

		SoilCOxisol	2.5	8.8	0.3	0.773	-14.7	19.8							
		SoilCULTisol	0.3	9.5	0.0	0.972	-18.3	19.0							
		ClimateBsh	7.9	8.0	1.0	0.325	-7.8	23.5							
		intrcpt	-0.6	5.8	-0.1	0.919	-11.9	10.7	Null	176.1	99.1	115.4	3345	0	
		Temperature_z	-2.8	2.5	-1.1	0.261	-7.8	2.1	Moderators	102.0	98.4	62.0	1897	0	42.1
		Rainfall_z	1.2	2.3	0.5	0.598	-3.3	5.7							
		Lat_z	-4.7	3.1	-1.5	0.127	-10.7	1.3							
		Time_span_z	-9.7	3.0	-3.2	0.001	-15.7	-3.7							
		Conv_Time_z	11.2	4.9	2.3	0.023	1.5	20.8							
		Long_z	-8.4	6.2	-1.4	0.175	-20.5	3.7							
	0 - 20	CSGrassland	9.5	5.1	1.9	0.061	-0.5	19.5							
		CSIAS	4.8	7.3	0.7	0.507	-9.4	19.0							
		CSSilviculture	-13.9	11.2	-1.2	0.211	-35.8	7.9							
		SoilCENTisol	-27.4	4.9	-5.6	0.000	-37.1	-17.8							
		SoilCInceptisols	3.0	10.3	0.3	0.772	-17.2	23.1							
		SoilCoxisol	-11.1	9.6	-1.2	0.247	-29.9	7.7							
		SoilCULTisol	-19.1	9.1	-2.1	0.035	-36.9	-1.4							
		SoilCVertisol	0.2	7.8	0.0	0.980	-15.1	15.5							
		ClimateBsh	4.6	7.0	0.7	0.508	-9.0	18.2							
		intrcpt	-0.6	5.8	-0.1	0.919	-11.9	10.7	Null	74.4	96.0	24.7	304	1E-52	
		Temperature_z	-2.8	2.5	-1.1	0.261	-7.8	2.1	Moderators	102.0	98.4	62.0	1897	0	-37.2
		Rainfall_z	1.2	2.3	0.5	0.598	-3.3	5.7							
		Lat_z	-4.7	3.1	-1.5	0.127	-10.7	1.3							
		Time_span_z	-9.7	3.0	-3.2	0.001	-15.7	-3.7							
		Conv_Time_z	11.2	4.9	2.3	0.023	1.5	20.8							
		Long_z	-8.4	6.2	-1.4	0.175	-20.5	3.7							
	0 - 30	CSGrassland	9.5	5.1	1.9	0.061	-0.5	19.5							
		CSIAS	4.8	7.3	0.7	0.507	-9.4	19.0							
		CSSilviculture	-13.9	11.2	-1.2	0.211	-35.8	7.9							
		SoilCENTisol	-27.4	4.9	-5.6	0.000	-37.1	-17.8							
		SoilCInceptisols	3.0	10.3	0.3	0.772	-17.2	23.1							
		SoilCoxisol	-11.1	9.6	-1.2	0.247	-29.9	7.7							
		SoilCULTisol	-19.1	9.1	-2.1	0.035	-36.9	-1.4							
		SoilCVertisol	0.2	7.8	0.0	0.980	-15.1	15.5							
		ClimateBsh	4.6	7.0	0.7	0.508	-9.0	18.2							
Pampa	0 - 10	intrcpt	-10.0	5.0	-2.0	0.04480	-19.8	-0.2	Null	16.0	90.8	10.9	255.2	5E-40	
		Temperature_z	-0.3	1.2	-0.2	0.81776	-2.7	2.1	Moderators	57.2	100.0	4555.5	1353.1	5E-240	-259

		Lat_z	4.2	1.2	3.5	0.00052	1.8	6.5							
		Conv_Time_z	-0.4	1.6	-0.2	0.80666	-3.5	2.7							
		Time_span_z	-0.7	1.1	-0.6	0.56140	-2.9	1.6							
		Long_z	-0.8	1.6	-0.5	0.62606	-3.9	2.4							
		CSConventional-Tillage	-1.3	5.4	-0.2	0.80687	-12.0	9.3							
		CSFlooded	0.5	9.9	0.1	0.95854	-18.9	20.0							
		CSIAS	-1.5	6.5	-0.2	0.82141	-14.2	11.3							
		CSNo-Tillage	3.4	5.2	0.7	0.50951	-6.8	13.6							
		CSSilviculture	23.6	8.0	2.9	0.00338	7.8	39.3							
		intrcpt	-10.0	5.0	-2.0	0.04480	-19.8	-0.2	Null	82.2	100.0	7684.2	5981.5	0	
		Temperature_z	-0.3	1.2	-0.2	0.81776	-2.7	2.1	Moderators	57.2	100.0	4555.5	1353.1	5E-240	30.4
		Lat_z	4.2	1.2	3.5	0.00052	1.8	6.5							
		Conv_Time_z	-0.4	1.6	-0.2	0.80666	-3.5	2.7							
		Time_span_z	-0.7	1.1	-0.6	0.56140	-2.9	1.6							
	0 - 20	Long_z	-0.8	1.6	-0.5	0.62606	-3.9	2.4							
		CSConventional-Tillage	-1.3	5.4	-0.2	0.80687	-12.0	9.3							
		CSFlooded	0.5	9.9	0.1	0.95854	-18.9	20.0							
		CSIAS	-1.5	6.5	-0.2	0.82141	-14.2	11.3							
		CSNo-Tillage	3.4	5.2	0.7	0.50951	-6.8	13.6							
		CSSilviculture	23.6	8.0	2.9	0.00338	7.8	39.3							
		intrcpt	-10.0	5.0	-2.0	0.04480	-19.8	-0.2	Null	55.4	97.7	43.1	447.0	1E-83	
		Temperature_z	-0.3	1.2	-0.2	0.81776	-2.7	2.1	Moderators	57.2	100.0	4555.5	1353	5E-240	-3.24
		Lat_z	4.2	1.2	3.5	0.00052	1.8	6.5							
		Conv_Time_z	-0.4	1.6	-0.2	0.80666	-3.5	2.7							
		Time_span_z	-0.7	1.1	-0.6	0.56140	-2.9	1.6							
	0 - 30	Long_z	-0.8	1.6	-0.5	0.62606	-3.9	2.4							
		CSConventional-Tillage	-1.3	5.4	-0.2	0.80687	-12.0	9.3							
		CSFlooded	0.5	9.9	0.1	0.95854	-18.9	20.0							
		CSIAS	-1.5	6.5	-0.2	0.82141	-14.2	11.3							
		CSNo-Tillage	3.4	5.2	0.7	0.50951	-6.8	13.6							
		CSSilviculture	23.6	8.0	2.9	0.00338	7.8	39.3							
		intrcpt	-10.0	5.0	-2.0	0.04480	-19.8	-0.2	Null	5	89	9	42	6E-08	
		Temperature_z	-0.3	1.2	-0.2	0.81776	-2.7	2.1	Moderators	57	100	4555	1353	5E-240	-1135
		Lat_z	4.2	1.2	3.5	0.00052	1.8	6.5							
		Conv_Time_z	-0.4	1.6	-0.2	0.80666	-3.5	2.7							
		Time_span_z	-0.7	1.1	-0.6	0.56140	-2.9	1.6							
	0 - 10	Long_z	-0.8	1.6	-0.5	0.62606	-3.9	2.4							
Pantanal															

	CSConventional-Tillage	-1.3	5.4	-0.2	0.80687	-12.0	9.3							
	CSFlooded	0.5	9.9	0.1	0.95854	-18.9	20.0							
	CSIAS	-1.5	6.5	-0.2	0.82141	-14.2	11.3							
	CSNo-Tillage	3.4	5.2	0.7	0.50951	-6.8	13.6							
	CSSilviculture	23.6	8.0	2.9	0.00338	7.8	39.3							
0 - 20	intrcpt	-10.0	5.0	-2.0	0.04480	-19.8	-0.2	Null	18	98	40	162	4E-33	
	Temperature_z	-0.3	1.2	-0.2	0.81776	-2.7	2.1	Moderators	57	100	4555	1353	5E-240	-216.1
	Lat_z	4.2	1.2	3.5	0.00052	1.8	6.5							
	Conv_Time_z	-0.4	1.6	-0.2	0.80666	-3.5	2.7							
	Time_span_z	-0.7	1.1	-0.6	0.56140	-2.9	1.6							
	Long_z	-0.8	1.6	-0.5	0.62606	-3.9	2.4							
	CSConventional-Tillage	-1.3	5.4	-0.2	0.80687	-12.0	9.3							
	CSFlooded	0.5	9.9	0.1	0.95854	-18.9	20.0							
	CSIAS	-1.5	6.5	-0.2	0.82141	-14.2	11.3							
	CSNo-Tillage	3.4	5.2	0.7	0.50951	-6.8	13.6							
0 - 30	CSSilviculture	23.6	8.0	2.9	0.00338	7.8	39.3							
	intrcpt	-10.0	5.0	-2.0	0.04480	-19.8	-0.2	Null	11	90	10	37	1E-07	
	Temperature_z	-0.3	1.2	-0.2	0.81776	-2.7	2.1	Moderators	57	100	4555	1353	5E-240	-439.3
	Lat_z	4.2	1.2	3.5	0.00052	1.8	6.5							
	Conv_Time_z	-0.4	1.6	-0.2	0.80666	-3.5	2.7							
	Time_span_z	-0.7	1.1	-0.6	0.56140	-2.9	1.6							
	Long_z	-0.8	1.6	-0.5	0.62606	-3.9	2.4							
	CSConventional-Tillage	-1.3	5.4	-0.2	0.80687	-12.0	9.3							
	CSFlooded	0.5	9.9	0.1	0.95854	-18.9	20.0							
	CSIAS	-1.5	6.5	-0.2	0.82141	-14.2	11.3							
CSNo-Tillage	3.4	5.2	0.7	0.50951	-6.8	13.6								
CSSilviculture	23.6	8.0	2.9	0.00338	7.8	39.3								

Biome = biome considered; **Soil layer (cm)** = depth of the soil layer evaluated. **Model** = fitted model type (random-effects = **Null** or mixed-effects = **Moderators**); τ^2 (**tau**²) = between-study variance (unexplained heterogeneity); **P** = proportion of total variability due to between-study heterogeneity; **H**² = ratio of total to sampling variance (alternative heterogeneity measure); **QE** = Q-test statistic for residual heterogeneity; **QEp** = p-value associated with the QE test; **R**² = proportion of heterogeneity explained by moderators.

Table 3. Summary of meta-analysis results stratified by biome and soil depth, presenting effect size estimates with confidence intervals (Random and Mixed models), heterogeneity statistics, and the explanatory influence of the moderator's latitude, mean annual temperature, and annual rainfall accumulation.

Biome	Soil layer (cm)	Model estimates and statistical parameters								Heterogeneity analysis			Bias and robustness analyses					
		Model	Term	Estimate	SE	Zval	Pval	CI.lb	CI.ub	τ^2	I ²	R ²	Egger test	Fail-Safe-N	Trim and Fill method			
															Estimate	CI.lb	CI.ub	k0
Cerrado	0 - 10	Null	intcpt	-4.0	0.43	-9.3	2.1E-20	-4.9	-3.2	49.5	97.7	-	0.07	246,311	-2.0	-2.9	-1.1	44
		Moderators	intcpt	-4.0	0.41	-9.8	1.2E-22	-4.8	-3.2	44.1	97.4	11.0						
			Lat	2.2	0.56	4.0	6.0E-05	1.2	3.3									
			Rainfall	0.7	0.47	1.5	1.5E-01	-0.2	1.6									
		Temperature	0.6	0.52	1.1	2.6E-01	-0.4	1.6										
	0 - 20	Null	intcpt	-7.3	0.67	-10.8	3.1E-27	-8.6	-5.9	94.7	98.5	-	0.17	451,232	-4.6	-6.0	-3.1	33
		Moderators	intcpt	-7.3	0.66	-11.1	2.0E-28	-8.6	-6.0	90.7	98.4	4.2						
			Lat	0.4	0.93	0.4	7.0E-01	-1.5	2.2									
			Rainfall	-0.4	0.68	-0.6	5.5E-01	-1.7	0.9									
		Temperature	1.9	0.93	2.0	4.2E-02	0.1	3.7										
	0 - 30	Null	intcpt	-5.8	0.63	-9.2	4.1E-20	-7.0	-4.6	51.7	97.9		0.05	139,618	-5.8	-7.0	-4.6	0
		Moderators	intcpt	-5.8	0.63	-9.2	4.5E-20	-7.0	-4.6	52.1	97.8	-0.8						
Lat			0.0	1.05	0.0	9.9E-01	-2.0	2.1										
Rainfall			0.9	0.64	1.3	1.8E-01	-0.4	2.1										
	Temperature	0.7	1.05	0.6	5.2E-01	-1.4	2.7											
Atlantic Forest	0 - 10	Null	intcpt	-5.5	0.80	-6.9	5.7E-12	-7.1	-3.9	61.2	99.1		0.86	124,470	-5.5	-7.1	-3.9	0
		Moderators	intcpt	-5.5	0.70	-7.8	5.0E-15	-6.9	-4.1	47.0	98.9	23.3						
			Lat	5.4	1.26	4.3	1.9E-05	2.9	7.8									
			Rainfall	1.3	0.90	1.5	1.5E-01	-0.5	3.1									
		Temperature	-1.0	1.19	-0.8	4.1E-01	-3.3	1.4										
	0 - 20	Null	intcpt	-7.6	1.03	-7.3	2.1E-13	-9.6	-5.6	90.1	97.1		0.82	54,057	-7.6	-9.6	-5.6	0
		Moderators	intcpt	-7.6	0.95	-8.1	6.9E-16	-9.5	-5.8	74.8	96.3	17.0						
			Lat	2.9	1.48	2.0	4.7E-02	0.0	5.8									
			Rainfall	4.7	1.21	3.9	1.0E-04	2.3	7.1									
		Temperature	1.5	1.66	0.9	3.5E-01	-1.7	4.8										
	0 - 30	Null	intcpt	-7.1	1.40	-5.0	4.6E-07	-9.8	-4.3	130.2	98.3		0.22	40,318	-11.4	-14.3	-8.4	16
		Moderators	intcpt	-7.0	1.36	-5.1	2.8E-07	-9.7	-4.3	123.0	98.1	5.5						
Lat			3.0	2.45	1.2	2.2E-01	-1.8	7.8										
Rainfall			0.6	1.59	0.4	6.9E-01	-2.5	3.8										
	Temperature	1.1	2.36	0.5	6.3E-01	-3.5	5.8											

Biome	Soil layer (cm)	Model	Term	Estimate	SE	Zval	Pval	CI.lb	CI.ub	τ^2	I ²	R ²	Egger test	Fail-Safe-N	Trim and Fill method				
															Estimate	CI.lb	CI.ub	k0	
Amazon	0 - 10	Null	intcpt	-4.0	0.69	-5.8	7.2E-09	-5.4	-2.7	4.2	52.9		0.22	741	-4.8	-6.2	-3.3	4	
		Moderators	intcpt	-4.1	0.65	-6.3	4.0E-10	-5.3	-2.8	2.9	44.2	30.6							
			Lat	-0.9	0.62	-1.4	1.5E-01	-2.1	0.3										
			Rainfall	0.9	0.79	1.2	2.4E-01	-0.6	2.5										
			Temperature	-0.7	0.79	-0.8	4.1E-01	-2.2	0.9										
	0 - 20	Null	intcpt	-4.9	0.93	-5.3	1.3E-07	-6.8	-3.1	22.9	90.2		0.99	3,022	-4.9	-6.8	-3.1	0	
		Moderators	intcpt	-4.8	0.86	-5.6	2.2E-08	-6.5	-3.1	18.6	87.8	18.5							
			Lat	-2.0	0.93	-2.2	3.1E-02	-3.8	-0.2										
			Rainfall	1.2	0.90	1.3	1.9E-01	-0.6	2.9										
			Temperature	-1.8	0.92	-2.0	5.1E-02	-3.6	0.0										
	0 - 30	Null	intcpt	-2.8	0.99	-2.9	4.3E-03	-4.8	-0.9	61.4	99.2		0.83	646,079	-2.8	-4.8	-0.9	0	
		Moderators	intcpt	-2.7	0.92	-3.0	3.0E-03	-4.5	-0.9	51.5	99.0	16.1							
Lat			-3.1	1.07	-2.9	3.4E-03	-5.2	-1.0											
Rainfall			2.4	0.91	2.7	7.3E-03	0.7	4.2											
Temperature			1.2	0.97	1.3	2.0E-01	-0.7	3.1											
Caatinga	0 - 10	Null	intcpt	-2.3	1.06	-2.1	3.4E-02	-4.3	-0.2	45.5	95.8		0.52	1,940	-2.3	-4.3	-0.2	0	
		Moderators	intcpt	-2.2	1.05	-2.1	3.5E-02	-4.3	-0.2	44.6	95.3	2.1							
			Lat	1.5	1.35	1.1	2.8E-01	-1.2	4.1										
			Rainfall	-1.7	1.04	-1.6	1.1E-01	-3.7	0.4										
			Temperature	-1.1	1.33	-0.8	4.0E-01	-3.7	1.5										
	0 - 20	Null	intcpt	-5.5	1.89	-2.9	3.8E-03	-9.2	-1.8	176.1	99.1		0.97	12,676	-5.5	-9.2	-1.8	0	
		Moderators	intcpt	-5.5	1.92	-2.9	4.1E-03	-9.3	-1.7	180.9	99.1	-2.7							
			Lat	-0.9	2.36	-0.4	7.1E-01	-5.5	3.7										
			Rainfall	-0.8	1.94	-0.4	6.7E-01	-4.6	3.0										
			Temperature	-1.8	2.36	-0.8	4.4E-01	-6.4	2.8										
	0 - 30	Null	intcpt	-4.3	1.95	-2.2	2.8E-02	-8.1	-0.5	74.4	96.0		0.73	991	-4.3	-8.1	-0.5	0	
		Moderators	intcpt	-4.3	2.09	-2.1	4.0E-02	-8.4	-0.2	86.1	96.4	-15.8							
Lat			0.0	3.08	0.0	9.9E-01	-6.1	6.0											
Rainfall			-0.6	2.21	-0.3	8.0E-01	-4.9	3.8											
Temperature			-1.7	3.12	-0.5	5.9E-01	-7.8	4.5											

-Continued on next page-

Model = fitted model type (null = random-effects; moderators = mixed-effects); τ^2 (tau²) = between-study variance (unexplained heterogeneity); **I²** = percentage of total variability due to between-study heterogeneity; **R²**

Biome	Soil layer (cm)	Model	Term	Estimate	SE	Zval	Pval	CI.lb	CI.ub	τ^2	I ²	R ²	Egger test	Fail-Safe-N	Trim and Fill method				
															Estimate	CI.lb	CI.ub	k0	
Pampa	0 - 10	Null	intrcpt	-5.4	0.83	-6.5	6.5E-11	-7.1	-3.8	16.0	90.8		0.06	4,753	-6.2	-7.9	-4.5	3	
		Moderators	intrcpt	-5.3	0.79	-6.8	1.4E-11	-6.9	-3.8	14.2	89.8	11.1							
			Lat	-1.8	1.12	-1.6	1.1E-01	-4.0	0.4										
			Rainfall	0.9	0.97	0.9	3.5E-01	-1.0	2.8										
	Temperature	1.4	0.99	1.4	1.6E-01	-0.6	3.3												
	0 - 20	Null	intrcpt	-7.4	1.07	-6.9	4.0E-12	-9.5	-5.3	82.2	100.0		0.46	63,154	-7.4	-9.5	-5.3	0	
		Moderators	intrcpt	-7.4	0.98	-7.5	4.5E-14	-9.3	-5.5	68.2	100.0	17.0							
			Lat	1.1	1.01	1.0	2.9E-01	-0.9	3.0										
			Rainfall	4.5	1.40	3.2	1.2E-03	1.8	7.3										
	Temperature	-0.8	1.44	-0.6	5.6E-01	-3.7	2.0												
	0 - 30	Null	intrcpt	-5.9	1.76	-3.3	8.2E-04	-9.3	-2.4	55.4	97.7		0.02	1,460	-4.3	-7.7	-0.9	3	
		Moderators	intrcpt	-6.0	1.72	-3.5	5.4E-04	-9.3	-2.6	53.2	97.8	4.0							
Lat			1.1	3.36	0.3	7.4E-01	-5.5	7.7											
Rainfall			-4.3	4.93	-0.9	3.9E-01	-13.9	5.4											
Temperature	3.6	5.65	0.6	5.2E-01	-7.5	14.7													
Pantanal	0 - 10	Null	intrcpt	-3.9	0.94	-4.2	3.2E-05	-5.8	-2.1	4.6	89.1		0.25	317	-3.9	-5.8	-2.1	0	
		Moderators	intrcpt	-4.0	0.84	-4.8	2.0E-06	-5.7	-2.4	3.5	89.2	23.5							
			Lat	0.5	1.24	0.4	6.7E-01	-1.9	3.0										
			Rainfall	-2.0	1.18	-1.7	9.5E-02	-4.3	0.3										
	Temperature	-	-	-	-	-	-												
	0 - 20	Null	intrcpt	-4.8	1.77	-2.7	6.6E-03	-8.3	-1.3	18.1	97.5		0.18	529	-4.8	-8.3	-1.3	0	
		Moderators	intrcpt	-4.9	1.56	-3.1	1.8E-03	-7.9	-1.8	13.9	97.4	23.1							
			Lat	-1.0	2.19	-0.5	6.4E-01	-5.3	3.3										
			Rainfall	-2.5	2.14	-1.2	2.4E-01	-6.7	1.7										
	Temperature	-	-	-	-	-	-												
	0 - 30	Null	intrcpt	-5.3	1.54	-3.5	5.2E-04	-8.3	-2.3	10.6	89.9		0.05	225	-5.3	-8.3	-2.3	0	
		Moderators	intrcpt	-5.3	1.44	-3.7	2.1E-04	-8.1	-2.5	9.1	88.0	14.0							
Lat			-2.3	1.67	-1.4	1.6E-01	-5.6	0.9											
Rainfall			-1.9	1.66	-1.1	2.5E-01	-5.1	1.4											
Temperature	-	-	-	-	-	-													

(%) = proportion of heterogeneity explained by moderators. **Egger test** = funnel plot asymmetry test for publication bias detection; **Fail-Safe-N** = number of additional null-effect studies required to nullify the observed significance; **Trim and Fill** = method that estimates and corrects the pooled effect by adjusting funnel asymmetry; **Estimate** = point estimate of the adjusted effect size; **CI.lb / CI.ub** = lower and upper bounds of the 95% confidence interval; **k0** = number of studies imputed by the Trim and Fill method

- Reviewer's comments in *blue*
- Replies in black
- A line numbering ("lines:") mentioned in the responses refers to the unmarked version of the manuscript.

REVIEWER COMMENTS

Reviewer #1 (Remarks to the Author):

Only very minor issues remain, primarily related to reference formatting. A few entries contain small typographical errors (e.g., references No. 21, 37, and 77). I recommend a final thorough check of the reference list for consistency (author names, journal titles, page ranges, and punctuation) before acceptance.

R: All references were corrected in the manuscript.

Reference

Line – 707, N° 21

Line – 746, N° 37

Line – 843, N° 77

Reviewer #3 (Remarks to the Author):

Line 174 - Soil carbon gap between native vegetation and agriculture

R: We thank the reviewer for the comment. We have added the missing space before the title (Line:176).

Line 198 - 020 cm

R: The typographical error in the indication '020 cm' has been corrected (Line:200).

Line 221 - "more moderate" what is a more adequate reduction?

R: We thank the reviewer for the suggestion. We revised the sentence to provide greater clarity and precision (lines: 222-224). The updated text now reads: "Under subtropical climates, similar losses were observed in the Atlantic Forest and Pampa (approximately -17.4%), while the Cerrado showed a smaller reduction (-9.8%)."

Line 268-269 - The Atlantic Forest... which increased to approximately -19% in areas older than 30 years. This pattern indicates a continuous process of degradation and highlights the high...

R: We adjusted this sentence in the text (lines: 267-271).

Line 277- tillage vs no tillage?

R: We thank the reviewer for this helpful observation. We added the information (lines: 278-279) on the age range of the agricultural system (up to 15 years) to the previous sentence and, based on the dataset for this age class in the Pampa, we also included the information on the management transition (tillage to no-tillage or integrated agricultural systems).

We present below the updated version of the sentence (lines: 276-279):

“The Pampa biome was the only one to show an increase (2%) in soil organic carbon stocks in systems up to 15 years old, although this increase was not statistically significant. This result may reflect transitions in management (tillage to no-tillage or Integrated agricultural systems) or the presence of residual C from the original vegetation.”

Line 286 - It can be rewritten as you mentioned a dual behaviour related to the same old age.

R: Thank you for your comment. We rewrote the sentence by adding the suggested information.

Below, we present the updated sentence (Lines: 286-290):

In contrast, the smaller gaps in the Amazon and Caatinga may reflect the predominance (~80%) of data from systems up to 30 years old, which exhibit a dual behaviour within this age class, as some results are close to zero while the 16-30-year group showing negative but non-significant results, indicating high variability and low C losses.

Line 298 - in a particular biome?

R: We adjusted this sentence in the text (lines: 298-301).

Line 418 - “agricultural intensification,” I complete understand the meaning of agricultural intensification but many researchers associate this strategy as agriculturización (change native to agricultural land use) So, I recommend you to define previously the meaning of this.

R: We appreciate the valuable comment, both from a methodological and a conceptual standpoint. To avoid any conceptual ambiguity associated with the term ‘agricultural intensification’, which may be interpreted by some readers as agriculturización (conversion of native areas to agricultural use), we added an explanatory note in the sentence where the term is first introduced (lines: 354–356). In this note, we clarify that the term should be understood within the scope of ‘sustainable intensification’, that is, increasing yields without adverse environmental impacts or additional conversion of non-agricultural lands (Pretty & Bharucha, 2014).

Below is the updated version of the sentence (lines: 353–357):

“These results revealed the benefits of agricultural intensification, understood within the framework of sustainable intensification, in which yields increase without adverse environmental impacts or the conversion of additional non-agricultural land (Pretty & Bharucha, 2014), mainly by biodiversification, to preserve/restore SOC_{stocks}.”

Line 491 - WebPlotDigitizer): land-use classes

R: The typological error has been corrected (Line:470).

Line 853 - Reference Plan Conversion.

R: All references were corrected in the manuscript.